# Rapid-Response Vector Surveillance and Emergency Control During the Largest West Nile Virus Outbreak in Southern Spain

**DOI:** 10.3390/insects16111100

**Published:** 2025-10-29

**Authors:** Mikel Alexander González, Carlos Barceló, Roberto Muriel, Juan Jesús Rodríguez, Eduardo Rodríguez, Jordi Figuerola, Daniel Bravo-Barriga

**Affiliations:** 1Departamento de Entomología y Sanidad Ambiental, Athisa Medio Ambiente (Grupo SASTI), C/Herrería-4, 41210 Sevilla, Spain; jjrodriguez@athisa.es (J.J.R.); erodriguez@athisa.es (E.R.); 2Estación Biológica de Doñana (EBD, CSIC), Avda, Américo Vespucio, 26, 41092 Sevilla, Spain; jordi@ebd.csic.es; 3Ciber de Epidemiología y Salud Pública (CIBERESP), Avda, Monforte de Lemos 3-5, 28029 Madrid, Spain; 4Applied Zoology and Animal Conservation Group, University of the Balearic Islands (ZAP—UIB), 07122 Palma, Spain; carlos.barcelo@uib.es; 5Tragsatec, Grupo Tragsa—SEPI, C/Parsi 5, nº8, 41016 Sevilla, Spain; rmuriel@tragsa.es; 6Department of Animal Health (Parasitology and Parasitic Diseases), Animal Health and Zoonosis Research Group (GISAZ), UIC Zoonoses and Emerging Diseases (ENZOEM), Faculty of Veterinary Medicine, Sanidad Animal Building, Rabanales Campus, University of Cordoba, 14014 Córdoba, Spain; dbravo.barriga@gmail.com

**Keywords:** adult control, *Bacillus thuringiensis*, *Culex pipiens*, *Culex perexiguus*, mosquito larval habitats, larval control, integrated vector management, public health response

## Abstract

**Simple Summary:**

In summer 2024, Spain experienced its largest outbreak of West Nile virus, a disease spread by mosquito bites that can cause serious illness in humans. Most cases occurred in small cities in the south-west of the country. In response, health authorities launched an emergency program to monitor and reduce mosquito populations. Our team worked across six municipalities with about 270,000 residents, where a third of all national cases were reported. Over four months (late July to mid-November 2024), we inspected 725 sites of seven aquatic habitat categories, including rice fields, ditches, and canals, to find mosquito larval spots. We identified eleven mosquito species, including two of epidemiological importance: *Culex pipiens* and *Culex perexiguus*. To control them, we treated affected areas with environmentally safe products that kill mosquito larvae and others that target adult mosquitoes. Significant reduction in larval abundance was observed after larvicide treatments but varied across treatment rounds. This study provides a more complete understanding of the larval mosquito fauna in Spain’s most important WNV hotspot and highlights the need for rapid, expert-led interventions to protect public health and prevent future outbreaks.

**Abstract:**

West Nile Virus (WNV) is an emerging arboviral threat in Europe, with rising incidence in Spain since 2004. In 2024, Spain experienced its largest outbreak, primarily in small urban areas of south-western regions. We report a subset of an emergency integrated vector management program, focusing on six municipalities accounting for one-third of all human WNV cases nationwide. Over four months, 725 potential larval sites were inspected during 4026 visits. Adult mosquitoes (*n* = 2553) were collected with suction traps, and immature stages (*n* = 4457) with dipper techniques, yielding 11 species. *Culex pipiens* s.l. was predominant, while *Cx. perexiguus*, though less abundant, was epidemiologically significant. Cytochrome Oxidase I (COI) gene phylogenetic analysis confirmed *Cx. perexiguus,* forming a distinct clade from *Cx. univittatus*. Immature mosquitoes were found in 18.6% of sites, especially irrigation canals, ditches, and backwaters near urban areas. Habitat differences in larval abundance were analyzed using generalized linear mixed models. Targeted larviciding with *Bacillus thuringiensis* var. *israelensis* (Bti) and focal adulticiding with cypermethrin totaled 259 interventions (70.4% larviciding, 29.6% adulticiding). A significant 63.9% reduction in larval abundance was observed after five consecutive Bti treatments, with some variation among treatment cycles (52.2–75.5%). Adult activity persisted into late autumn. This study provides the first comprehensive characterization of larval mosquitoes in Spain’s main WNV hotspot, highlighting the need for rapid, coordinated expert interventions and extended seasonal control to prevent future outbreaks.

## 1. Introduction

Mosquitoes pose a global threat as vectors of pathogens of public health and veterinary importance, including viruses such as dengue, Zika, and West Nile virus (WNV), protozoans (*Plasmodium* spp.) and filarial nematodes, among others [1,2]. Land-use changes, globalization, invasive species and climate change are the main factors that affect mosquito distribution and the frequency of outbreaks of vector-borne diseases [3,4,5].

WNV is a globally distributed zoonotic disease whose spread has increased over the last decade, posing a global challenge to public health [6]. The virus is primarily maintained in an enzootic transmission cycle between mosquitoes and birds, but it can become epizootic and infect humans and other mammals, occasionally leading to severe neurological disease. First identified in Uganda in 1937, WNV has expanded its geographic range, causing recurring outbreaks across Africa, the Middle East, Europe, and the Americas [7,8,9]. In Europe, significant outbreaks have occurred in Greece, Italy, and Serbia [10]. In Spain, the virus has been circulating since 2003, with documented transmission among birds, mosquitoes, equids, and humans [11,12,13]. The country has also seen a notable rise in human cases, particularly in the southern regions, with major outbreaks in 2020 (77 cases, 8 deaths) and 2024 (158 cases, 20 deaths) [14,15]. Experience gained from the 2020 outbreak guided the implementation of improved surveillance protocols, which were further refined during 2024 and underpin the current emergency vector management effort.

In Europe, the main WNV vector species are *Culex pipiens*, *Culex perexiguus*, *Culex univitattus*, *Culex modestus*, and *Culex torrentium* [16,17] but WNV has also been isolated from other *Culex* and *Aedes* species, and recent competence studies also highlighted the potential of other species [18,19,20]. In Spain, the most problematic mosquito species regarding WNV transmission are *Cx. pipiens* and *Cx. perexiguus* [11,21]. While *Cx. perexiguus* is considered the main vector of WNV among birds in natural and agricultural areas, its role in urban environments still requires further investigation. Agricultural landscapes surrounding urban areas can act as reservoirs where enzootic transmission cycles are amplified, and mosquitoes or infected birds can subsequently move into urban nuclei. Once the cycle is driven by *Cx. perexiguus* occurs in nearby villages, *Cx. pipiens* may act as a bridge vector, facilitating transmission from birds to humans [22]. Targeted surveillance and control of *Cx. perexiguus* populations therefore appear to be the most effective measures to reduce WNV amplification [11]. The growing recognition of *Cx. perexiguus* as the primary WNV vector, surpassing *Cx. pipiens*, is supported by differences in feeding behavior, vector competence, and ecological preferences that directly influence the dynamics of WNV transmission [23]. The province of Seville hosts natural wetlands and extensive irrigated landscapes, such as rice fields and irrigation canals, which are ideal larval habitats for mosquito vectors. However, little is known about species-specific larval habitats in these systems. This study addresses this gap by characterizing larval habitats and associating them with targeted emergency control actions, providing new large-scale evidence that integrates operational interventions with molecular confirmation of *Cx. perexiguus*.

The recurrence and intensity of WNV outbreaks underscore the urgent need for early detection and timely vector control measures. Effective surveillance provides critical data on vector abundance, species composition, and seasonal activity, allowing for targeted interventions and better risk assessment [24]. At the same time, Integrated Vector Management (IVM) has emerged as a key strategy to reduce mosquito populations and interrupt transmission. IVM includes vector monitoring, identification of the problem and a combination of environmental, biological, and chemical control methods, supported by public education and multisectorial coordination. Several studies have shown that larviciding in standing water habitats and targeted adulticide applications during high-risk periods may reduce mosquito populations and help mitigate WNV transmission [21,25,26]. However, effectiveness varies depending on the techniques and formulations applied. Although the application of these methods in Spain has apparently yielded promising results, little supporting scientific evidence has been reported so far.

Given the increasing public health threat posed by WNV and other emerging arboviruses, it is critical to strengthen surveillance and response systems. In this study, we present an integrated operational response to the 2024 WNV outbreak in southern Spain, focusing on entomological monitoring and coordinated vector control actions. This study aimed to generate robust insights to guide the design and implementation of more effective WNV prevention strategies and to strengthen broader preparedness efforts for vector-borne diseases in a changing environment. The integrative surveillance approach also provided detailed information on habitat species composition, distribution, and relative abundance of mosquito species in WNV-affected areas, thereby supporting scientifically informed vector control interventions. We hypothesized that surveillance and control activities focused around inhabited areas would be essential to effectively reduce vector populations and protect humans.

## 2. Material and Methods

### 2.1. Study Area

This study was conducted as part of a vector surveillance and control program implemented by Athisa Medio Ambiente (Grupo SASTI) under a contract with Diputación de Sevilla, a supramunicipal public administration in south-west Spain. The program was initiated in response to a substantial increase in human WNV cases during the early summer of 2024 and became operational in late July. The study area was in the Bajo Guadalquivir district of southern Spain and includes six principal municipalities: Dos Hermanas (DH), Las Cabezas de San Juan (LC), Utrera (UT), Los Palacios y Villafranca (PV), Alcalá de Guadaíra (AG), and Lebrija (LE), with a total combined population of approximately 270,000 inhabitants (Figure 1). Collectively, these municipalities accounted for 42 confirmed WNV cases in 2024, representing roughly one third of all reported cases in Spain. Following the guidelines established by regional health authorities, surveillance and control activities focused on peri-urban areas are conducted in adjacent zones to urban centers, where vector developmental habitats are commonly found. Climatically, the area has a hot-summer Mediterranean climate (Csa), with very hot, dry summers (mean August temperature of around 28–30 °C, with typical daytime maxima of about 40 °C) and rainfall concentrated from autumn to early spring (annual around 500–550 mm, peak in November–December, minimal in June–August).

### 2.2. Mosquito Sampling Strategy

Fieldwork was preceded by a detailed mapping of the study area using QGIS software (v. 3.34.9-Prizren) to identify potential mosquito larval habitats and high-risk zones. A 1500 m buffer zone was delineated around inhabited areas of the six selected municipalities for larval surveillance, in line with the Andalusian health authority’s instructions [27]. Following this, a minimum of two survey teams were deployed to systematically characterize aquatic larval sites across the municipalities.

A larval mosquito sampling campaign was conducted across the previously described area, covering a broad range of aquatic habitats (Figure 1). At each site, immature mosquito stages were sampled using a standardized dipping method consisting of 10 dippers per habitat [28,29], yielding approximately 4 l of water per site. The number of larvae collected was counted and the average number per dipper was calculated. All sampling points were georeferenced using GPS.

Selected sampling sites were revisited regularly throughout the study period. To estimate species richness and larval abundance, a subset of 63 sampling sites, representing approximately 46% of the total positive dataset samples, was preserved in ethanol and transported to the laboratory for taxonomic identification. Aquatic habitats were classified into seven major categories: (i) artificial ponds, often created for irrigation, livestock, or ornamental purposes; (ii) canals and ditches, which are irrigation or drainage channels that may hold stagnant water; (iii) drainage systems, including underground or surface infrastructures similar to “storm water catch basins” that can accumulate standing water; (iv) rice fields, which provide extensive and seasonally flooded habitats; (v) animal drinking troughs, where water may stagnate if not regularly maintained; (vi) natural watercourses and backwaters, where slow-moving sections or small branches of rivers allow larval development; and (vii) areas of waterlogging, referring to low-lying zones where rainfall or irrigation water accumulates and persists for extended periods. Sampling intensity varied among habitat categories.

Adult mosquito populations were monitored using five permanent BG-Sentinel traps (Biogents, Regensburg, Germany) equipped with attractant lures (Mosquito lure), strategically placed in urban areas of the municipality of Dos Hermanas (Figure 1). Trap contents were collected on a weekly basis from mid-July to mid-November 2024.

### 2.3. Mosquito Identification and Phylogenetic Analysis

Both adult and immature mosquito specimens were identified to species level using morphological keys, following the taxonomic criteria established by Becker et al. [30]. Fourteen *Cx. perexiguus* individuals were analyzed by molecular barcoding (Appendix A) to confirm their distinction from morphologically similar species *Cx. univittatus* [31]. Sequences of the mitochondrial cytochrome c oxidase subunit I (COI) gene were aligned using MAFFT v7.450 [32] with default parameters and manually curated to correct potential misalignments and remove ambiguous regions. Phylogenetic analyses were performed using the maximum likelihood (ML) approach implemented in IQ-TREE v1.6.12 [33], executed on a Linux 64-bit system. Further methodological details and supporting data related to the phylogenetic analyses are available in Appendix A.

Final tree visualization and editing were performed using FigTree v1.4.4 (accessed on 9 September 2025; http://tree.bio.ed.ac.uk/software/figtree/).

### 2.4. Statistical Analysis of Habitat Differences in Larval Abundance

To assess whether mosquito larvae abundance differed among habitat types, we fitted a generalized linear mixed model (GLMM) with a negative binomial distribution (NB2 parameterization) to account for overdispersion typically observed in count data. As only samples with larval presence were analyzed, zero counts were absent; thus, zero-inflation was not assessed. Instead, model fit was evaluated by comparing Poisson and negative binomial GLMMs, the latter providing an appropriate fit without overdispersion. The total number of larvae per sample was included as the response variable, while habitat category was included as a fixed-effect factor. The habitat level “water through” was excluded from the analysis due to insufficient data. The sampling site identity was incorporated as a random intercept to control for pseudo-replication resulting from repeated measures at the same locations. Models were fitted using the glmmTMB function from the glmmTMB package in R (Version 4.5.1). The model fit was evaluated using Pearson’s χ^2^ dispersion statistic and the significance of the fixed effect of habitat. In addition, model diagnostics were conducted by visually inspecting the scaled residuals and testing model assumptions using the DHARMa package in R (Version 0.4.6) [34] (Appendix A). The proportion of variance explained by the model was estimated following [35] using the r2_nakagawa function from the performance package. We report both the marginal R^2^ (variance explained by fixed effects) and the conditional R^2^ (variance explained by both fixed and random effects). When the fixed factor was significant (*p* < 0.05), estimated marginal means and pairwise post-hoc comparisons with Tukey’s (HSD) adjustment were computed using the emmeans package. Graphs and data of larval density were **analysed** using log-transformed densities to account for skewness. Geometric means were used to arrange habitats according to their relative productivity. All statistical analyses were performed by R [36].

### 2.5. Mosquito Control Interventions

We followed the same strategy as with surveillance: a 1500 m buffer zone was delineated around inhabited areas of the six selected municipalities for biological control and adulticide treatments. Targeted larval control interventions were carried out at all sites where larvae were detected. Treatment was triggered even by the presence of a single larva. Control measures employed formulations based on *Bacillus thuringiensis* var. *israelensis* (Bti), specifically Aquabac XT (Becker Microbial Products, Inc., Miami, FL, USA) or Vectobac^®^ 12 AS (Kenogard, Miami, FL, USA), depending on product availability. Products were applied at 1.5–2.5 L/ha after being previously diluted in water (suspension of 1% concentration), following the manufacturers’ recommendations. The dosage was increased by 50% only in cases where a high amount of organic matter was observed (e.g., effluent discharge). Follow-up inspections were typically performed within one week after larvicidal treatments (mean: 6.2 days, 95% CI: 5.5–6.9). When larvae persisted, subsequent interventions were scheduled within an average of 11.4 days between treatments (95% CI: 9.5–13.3), depending on larval persistence and logistic schedule. The choice of application method was tailored to the specific nature of each larval site: manual application (backpack) was used in small or isolated water bodies; vehicle-mounted sprayer cannons were deployed in large, open, and flooded areas; and drone-assisted dispersal was implemented in inaccessible or environmentally sensitive zones where ground access was limited.

Targeted adulticide treatments were applied within the buffer zones around populated areas, with the aim of reducing mosquito–human contact. These zones were defined through ground-based surveys that identified vegetated areas and paths with high potential for mosquito resting sites. Subsequently, complementary adulticide treatments were applied using pyrethroid-based products: Massocide^®^ Ciper 100 (Onatti Agr Est. Comercial Química Massó, S.A., Barcelona, Spain) or Fortex Next (Pestnet, Valencia, Spain). The insecticidal solution was prepared at a concentration of approximately 1% of the formulated product in water. Residual ground applications were carried out using vehicle-mounted low-volume spraying equipment, calibrated to deliver between 0.5–1.0 L/ha of the diluted product, in accordance with manufacturers’ recommendations and regional guidelines. Treatments were applied directly to vegetation and other resting sites surrounding populated areas, creating barrier zones to prolong adulticidal activity and reduce mosquito–human contact. Reapplications were conducted when adult mosquito densities remained high, particularly in areas with continued risk of transmission.

All vector control operations adhered strictly to established national and regional protocols regarding biocide use, occupational safety, and environmental protection, ensuring responsible and effective application throughout the intervention period.

### 2.6. Evaluation of Control Effectiveness

To evaluate the effectiveness of larval control interventions, we performed paired comparisons between pre-treatment and post-treatment at the positive sampling sites over five treatment rounds containing immature stages of mosquitoes. The study did not include an untreated control group. For the analyses, a treatment round was defined as each pre–post pair of observations at a positive site during de pre-treatment. The pre-treatment value corresponded to a visit with larvae (>0), and the subsequent inspection at the same site was considered the post-treatment value. If larvae persisted after treatment, that post value was treated as the new pre-treatment for the following round. Although more than five treatment rounds occurred in the dataset, only the first five were analyzed in detail, as subsequent rounds involved very few sites and did not provide sufficient sample sizes for robust statistical testing. The mean interval between pre- and post-treatment inspections at sites with ≥2 larval detections was approximately one week, reflecting the monitoring frequency. Sites with only a single positive detection or where the follow-up inspection occurred too late (more than 1 month) were excluded from this calculation, as no valid interval was available. For each treatment event, the average number of larvae before and after treatment was compared with a non-parametric Wilcoxon signed-rank test, allowing an evidence-based evaluation of treatment effectiveness. Treatment outcomes were also classified qualitatively as follows: treatments were considered effective when no larvae were detected in the subsequent inspection (larval presence = 0), ineffective when larvae were still present (larval presence > 0), and unclassified when no follow-up visit had been recorded. All statistical analyses were performed by R [36].

## 3. Results

Entomological surveillance of both adult and immature mosquito fauna allowed the identification of eleven mosquito species. The abundance of each species showed marked differences between larval and adult collections. The study was performed between 24 July and 15 November in 725 sampling points across the six municipalities (DH = 146, LC = 64, UT = 215, PV = 102; AG = 116, and LE = 86) with a sampling effort of 4026 site inspections (mean of 5.5 visits/site). Of the total inspected sites, 137 distinct sampling points (18.9%) tested positive for mosquito immature stages at any time during the study period.

### 3.1. Larvae Collections and Habitat Differences

The estimated density of larvae collection in the 4026 larval inspection points is represented in Figure 1. Spatial analysis revealed marked heterogeneity in larval densities. The highest concentrations (>300 larvae per sampling point) were located mainly in peri-urban areas south and southeast of the Seville province. The observed spatial distribution identifies priority foci for targeted vector-control interventions and temporally persistent or poorly managed aquatic habitat settlements. The map represents the total larval density (averages) per sampling site over the entire study period (Figure 1).

Up to 4457 specimens encompassing nine species were identified based on collections from 63 subsamples allocated in the following habitats: artificial ponds (*n* = 3), canals and ditches (*n* = 22), drainage systems (*n* = 3), rice fields (*n* = 3), water trough (*n* = 1), water course (*n* = 16), and flooded ground (*n* = 15).

*Culex pipiens* s.l. was the most abundant species, representing 90.8% of all collected specimens, with a mean abundance of 62.2 individuals per positive sampling site (Figure 2) and was dominant across the seven habitat types (Figure 3 and Appendix A). Although its abundance was relatively low, *Cx. perexiguus* was detected in almost all surveyed habitats except in artificial ponds and water troughs (Figure 3). *Culex theileri* was the second most abundant species in rice fields and ditch canals, while *Aedes* was frequently detected in flooded areas and artificial ponds (Figure 3 and Appendix A).

The most productive larval habitats were ditches and irrigation canals, which accounted for over 50% of all collected larvae, followed by artificial ponds and blackwater environments (Appendix A). Mean larval density per habitat (49.9) and median (7) showed clear variation: 18.5% of sites hosted fewer than 5 larvae, 62.3% between 5 and 50, and 19.2% exceeded 50 individuals. Larval densities varied markedly across habitats. Artificial ponds and ditch canals showed the highest geometric means and the widest variability, while rice fields and water troughs displayed consistently low values. Habitats linked to drainage systems and flooding (ditch canals, blackwater watercourses, flooded grounds) sustained moderate densities under strong sampling effort. Overall, the results highlight the disproportionate contribution of specific aquatic habitats, particularly anthropogenic water bodies, to mosquito larval development.

A negative binomial GLMM provided the best fit to the data (χ^2^ Pearson dispersion = 0.540), whereas a Poisson distribution showed strong overdispersion (χ^2^ Pearson dispersion = 45). The model revealed a marginally significant global effect of habitat on larval abundance (Wald χ^2^ = 10.098, df = 6, *p* = 0.073). Variance partitioning indicated that the model explained a substantial portion of the total variability (conditional R^2^ = 0.735), although most of it was attributed to random site-level differences, while the habitat effect accounted for a smaller fraction (marginal R^2^ = 0.138). Although visual inspection suggested higher larval abundance in artificial ponds compared with other habitats, post-hoc pairwise tests only indicated a significantly higher abundance in artificial ponds relative to water courses with blackwater (rate ratio = 14.15, SE = 13.10, z = 2.87, *p*-adjusted = 0.047) and a marginally higher abundance in comparison with flooded grounds (rate ratio = 11.99, SE = 11.20, z = 2.67, *p*-adjusted = 0.082) and rice fields (rate ratio = 22.27, SE = 26.00, z = 2.66, *p*-adjusted = 0.083) (Appendix A).

### 3.2. Barcoding and Phylogenetic Analysis

All the analyzed *Cx perexiguus/Cx. univitttatus* (*n* = 14) were confirmed as *Cx. perexiguus* and 8 *Cx. perexiguus* high-quality COI sequences were successfully generated and deposited in the DNA Data Bank of Japan (accessed on 9 September 2025; DDBJ: https://www.ddbj.nig.ac.jp/index-e.html) (Appendix A). The phylogenetic analysis based on the mitochondrial COI gene revealed a clear separation between our samples (LC883882-89) and *Cx. univittatus* reference sequences (Figure 4). Our samples consistently clustered within a well-supported clade alongside other sequences previously identified as *Cx. perexiguus*, confirming their correct species assignment.

### 3.3. Adult Collections

A total of 2553 specimens belonging to nine species were captured using BG-Sentinel traps (Appendix A). The primary vectors of WNV and other arboviruses *Cx. perexiguus* (*n* = 1154, 45.2%)*, Aedes albopictus* (*n* = 606, 23.7%), and *Cx. pipiens* s.l. (*n* = 378, 14.8%) were the most frequently detected species. The remaining species composition is shown in Appendix A. Adult mosquito activity peaked in August, with sustained high densities during September and October. Notably, adult specimens were detected throughout the year (Appendix A).

### 3.4. Control Assessment

An integrated management strategy was implemented at 137 larva-positive sites, resulting in a total of 484 targeted interventions: 70.3% against immature stages and 29.6% targeting adult mosquitoes. Larvicidal treatments produced consistent reductions in larval densities across most treatment rounds (Figure 5).

Overall, larviciding was effective in 63.9% of treatments (*n* = 100), reducing mean larval densities from 33.8 (pre-treatment) to 12.2 (post-treatment) larvae per site. In rounds with significant effects, many sites reached zero larvae post-treatment, while others showed partial persistence, consistent with overall reductions in larval abundance. Hodges–Lehmann estimates confirmed effect sizes between 5 and 10 larvae per site (95% CI: 3–40). Significant pre–post differences were detected in rounds 1, 3, and 5 (Wilcoxon signed-rank test, *p* ≤ 0.05), while no significant effect was detected in rounds 2 (*p* ≥ 0.05) and was nearly significant in round 4 (*p* = 0.052), although the number of samples in these rounds was relatively lower, which may have reduced statistical power (Figure 5).

When outcomes were classified as effective (no larvae detected post-treatment) or ineffective (larvae still present), analyses indicated complete suppression in 52.2–75.5% of cases, while the remainder showed partial reductions or persistence. These findings indicate that larvicidal interventions can reduce larval populations under field conditions, but the level of effectiveness varied notably across treatment rounds.

## 4. Discussion

Our study presents a comprehensive analysis of an intensive and sustained campaign combining both entomological surveillance and vector control interventions to mitigate the impact of mosquito species of public health importance. We identified the most critical larval habitats for key WNV vectors *Cx. pipiens* and *Cx. perexiguus,* as well as for several other mosquito species, within one of the regions most affected by WNV in Spain.

Although the *p*-value was marginal, the consistent pattern of higher larval abundance in canals and ditches suggests that this result likely reflects genuine habitat differences rather than low statistical power. The well-known *Cx. pipiens* s.l. exhibited a wide distribution across diverse habitat types, reflecting its ecological plasticity and capacity to thrive in both natural and anthropogenic settings [28,30,37]. In contrast, *Cx. perexiguus* was more prevalent in water courses and rice field habitats, and to a lesser extent in drainage systems and other flooded environments. These findings are noteworthy, as the larval ecology of *Cx. perexiguus* remains poorly studied in Europe. Our results, in line with studies from Spain [38], indicate a predominance in rice fields, although immature stages have been reported in a broad range of stagnant waters, from clean to moderately polluted habitats such as swamps, ponds, slow-flowing streams and pools. This species also shows tolerance to salinity [30,39,40], underscoring its adaptability to diverse aquatic environments. The other two species, also highlighted as suspected WNV vectors, *Cx. laticinctus* and *Cx. modestus,* were found mainly in drainage systems and ditch canals, respectively.

The identification of *Cx. perexiguus* can be challenging due to its close morphological similarity with *Cx. univittatus*, a species with which it may co-occur in certain regions [41]. In our work, we analyzed a few larval specimens from four different habitat categories to confirm their taxonomic identity through molecular analysis. COI sequences consistently grouped within a strongly supported *Cx. perexiguus* clade while *Cx. univittatus* formed a distinct clade, underscoring the separation between the two taxa. This confirms that the collected individuals correspond to *Cx. perexiguus*, a competent vector of WNV, as opposed to *Cx. univittatus*, which is apparently not considered a relevant vector in the region studied. Interestingly, *Cx. univittatus* has been widely detected in Portugal, as confirmed by surveillance data from REVIVE (2024) [42], while studies in Extremadura indicate the presence of both *Cx. perexiguus* and *Cx. univittatus* [42], suggesting potential areas of sympatry between both species. Nevertheless, *Cx. univittatus* generally occurs as a minor species in abundance. This regional variability highlights the importance of conducting molecular analyses to avoid misidentification, as the epidemiological implications could differ depending on which species is present. In our study area, however, the available evidence and previous studies in the area [11] indicate that agricultural fields and waterlogged habitats appear to exclusively host *Cx. perexiguus*. This clear phylogenetic segregation rules out potential taxonomic misidentification or the presence of cryptic species in our dataset and reinforces the initial morphological identification of the specimens as *Cx. perexiguus.*

For several decades, Bti has been widely applied in mosquito control programs across Europe to mitigate nuisance, and it is generally regarded as an environmentally safe, effective, and highly target-specific biocide [43]. The use of Bti as a larvicide has demonstrated highly effective control and persistence, usually between 4 and 8 weeks depending on the formulation, against *Culex* mosquitoes under laboratory, semi-field and field conditions [44,45,46,47]. However, evidence from real-world extensive applications is comparatively limited, and effectiveness under field conditions may vary significantly due to environmental and operational factors [48]. Our study provides one of the first large-scale evaluations of Bti applications against WNV vectors in Spain, addressing a knowledge gap where, despite the routine use of larvicides in recent outbreaks, operational evidence has been scarce.

Our results support the ability of Bti in reducing larval mosquito populations in treated areas, although differences were observed among treatment cycles, and some treated habitats showed rapid recolonization after intervention. Such variability in larviciding outcomes is influenced by multiple interacting factors, including the target mosquito species, its developmental phase, water temperature, sunlight exposure, larval density, the presence of organic matter, the physiochemical properties of each habitat, and the specific formulation of the Bti product used, among others [44,45,46,47]. In irrigation ditches, drainage channels, and rice fields, even low water flow can reduce residual activity through dilution or displacement of the larvicide [49]. Furthermore, high summer temperatures (35–45 °C) likely accelerate degradation, while rainfall and flow fluctuations contribute to reduced persistence. Altogether, these conditions explain uneven performance (i.e., second round low effectiveness) and highlight the need for habitat-tailored doses and shorter retreatment intervals. The lack of significance in round 4 likely reflects reduced sample size and variability rather than an absence of a treatment effect. Field studies have consistently recommended the reapplication of larvicides every 8 to 10 days to align with the typical developmental cycle of mosquitoes and to maintain effective population suppression [26]. Our findings suggest that in unstable or semi-permanent habitats, larvicide applications may need to be substantially more frequent than originally planned, requiring adaptive management strategies tailored to local ecological conditions. To manage this variability and reduce the risk of diminished effectiveness over time, programs should incorporate routine susceptibility monitoring and mode-of-action rotation. When rapid recolonization or suboptimal effectiveness is observed, operators should alternate Bti with alternative larvicides matched to habitat conditions: *Lysinibacillus sphaericus* (Bs) or Bti–Bs combinations in organically rich or polluted waters. Applying these tools in rotation or mosaic, alongside product quality control and water-quality checks, can help maintain larvicidal performance while mitigating resistance selection and environmental impact.

Adult mosquito collections confirmed the high abundance of *Cx. perexiguus* in urban areas as well, as was already recorded during the 2020 outbreaks [14]. In urban areas, *Cx. pipiens* is typically the dominant species [28] but the relative proximity of *Cx. perexiguus* preferred habitats to urban areas may explain its high abundance inside cities. The dominance of this species in these collections raises the possibility that individuals of this species are migrating to urban environments from surrounding larval habitats, such as irrigated agricultural zones and rice fields and temporarily outnumbering *Cx. pipiens* populations that breed within the urban foci. From a public-health standpoint, this dynamic calls for integrating peri-urban and agricultural belts into urban surveillance and control plans, prioritizing targeted actions along the urban–agriculture interface, coordinating across municipalities, and aligning risk communication with periods of heightened movement. Moreover, our findings demonstrate that adult mosquito activity persists well into late autumn, underscoring the need to extend entomological surveillance beyond the traditional summer peak to enable earlier detection of viral circulation and timely implementation of control measures in subsequent seasons.

A key limitation of the study was the inability to assess mosquito population reduction at regular and standardized intervals, which restricted the robustness of statistical analyses. This limitation arose from the need to prioritize spatial coverage across all study sites, which constrained the ability to maintain consistent sampling frequencies. The absence of untreated control areas reflected the emergency context, in which rapid vector suppression took precedence over experimental design. The absence of untreated control areas reflected the emergency context, in which rapid vector suppression took precedence over experimental design. Despite these limitations, our results underscore the variability of larviciding outcomes and the importance of integrating environmental and operational factors into control planning. Future programs should coordinate entomological and epidemiological surveillance to enable spatiotemporal analyses and more robust assessments. Engagement and communication with affected communities should also be strengthened to support rapid and sustainable interventions.

Although our study did not assess the effectiveness of adulticide applications, previous operational experiences, including large-scale interventions in European rice-growing areas, have shown that perimeter spraying with pyrethroid-based products can lead to short-term reductions in adult *Culex* populations [21]. These interventions, when guided by entomological surveillance, may serve as a valuable complement during periods of elevated nuisance or virus circulation. However, their use should be considered a reactive and targeted measure, always integrated within a broader framework of sustained larval control to ensure long-term effectiveness and minimize environmental impact. Given the current context of insecticide resistance emerging in *Aedes* and *Culex* species in Europe [50,51], careful monitoring of susceptibility should also be incorporated into future control programs. Because reliable power and secure, authorized sites for trap deployment were only available in one municipality during the rapid-response phase, adult abundance data are limited to that municipality, which may constrain extrapolation to the wider study area.

## 5. Conclusions

Mosquito vectors such as *Cx. perexiguus* and *Cx. pipiens* s.l. were detected in a wide range of habitats, indicating that no aquatic habitat should be excluded. Larval control with Bti showed moderate and variable reductions in mosquito populations across habitats, suggesting that this approach can contribute to vector management when applied earlier, more broadly, and for longer durations. Comprehensive and adaptive larval control strategies are therefore recommended to reduce vector abundance and transmission risk. These findings provide useful operational evidence for public health authorities in endemic areas and underscore the value of integrating ecological, molecular, and operational approaches for outbreak response. These findings support the integration of entomological surveillance within regional One Health preparedness frameworks.

## Figures and Tables

**Figure 1 insects-16-01100-f001:**
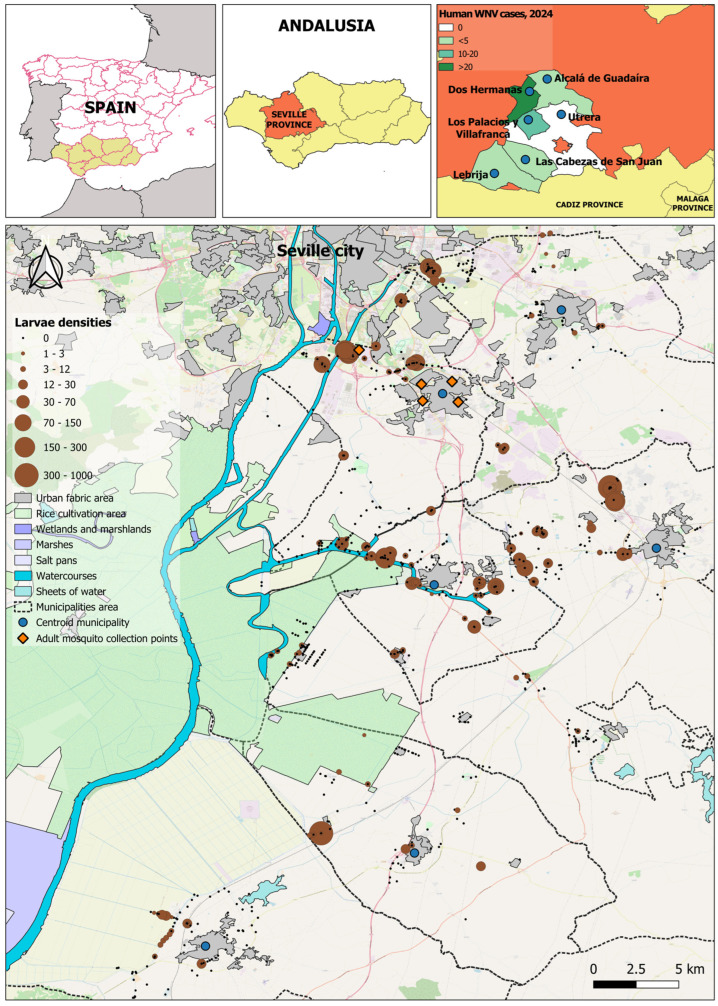
Spatial distribution of mosquito larval densities in the metropolitan area of Seville between late July and mid-November 2024. Brown circles represent larval sampling sites, with circle size scaled into eight classes according to the natural breaks (Jenks) classification method, indicating the average number of larvae detected per sampling point. Blue circles indicate sampled municipalities and orange diamonds indicate adult sampling points. The map includes a north arrow and a scale bar (0–5 km) for reference. Map background: OpenStreetMap.

**Figure 2 insects-16-01100-f002:**
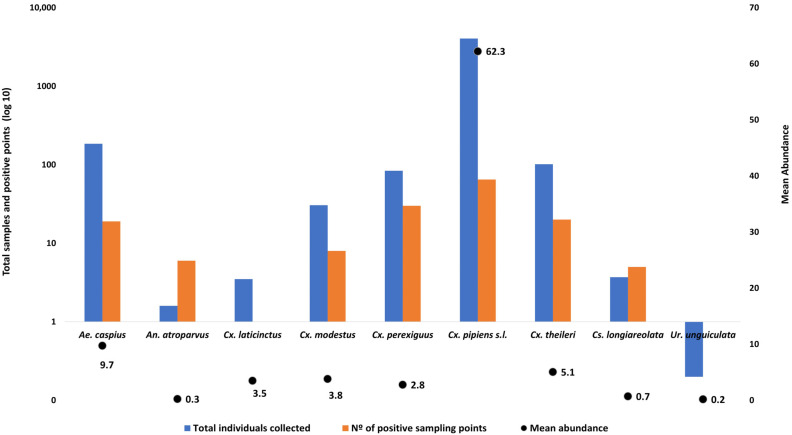
Total number of adult mosquito individuals collected across all sampling sites (blue bars, log_10_ scale), number of positive sampling points where each species was detected (orange bars, log_10_ scale), and mean abundance per positive site (black dots) during the surveillance period. Mean abundance was calculated as the average number of individuals per species in those sites where the species was present. Data correspond to adult mosquito collections from the study area (Sevilla province, southern Spain).

**Figure 3 insects-16-01100-f003:**
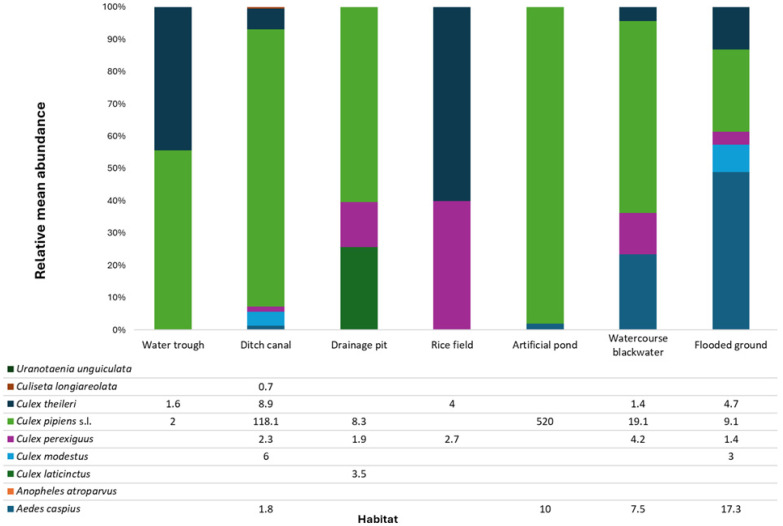
Mean relative abundance (%) of immature mosquitoes recorded in seven different habitat categories in the study area (Seville province, southern Spain). Values in the stacked bars were calculated as the mean proportion of individuals per species across all positive sampling sites within each habitat category. Only species representing more than 0.5% of the total catch in a given habitat are shown.

**Figure 4 insects-16-01100-f004:**
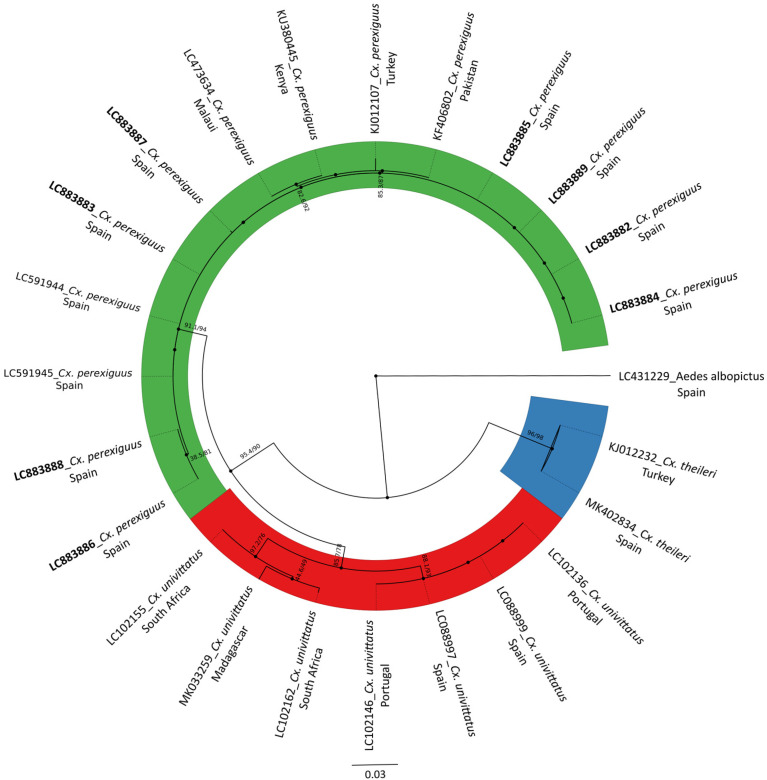
Circular phylogenetic tree of *Culex* spp. based on 24 mitochondrial COI gene sequences (658 bp), inferred using the Maximum Likelihood (ML) method in IQ-TREE under the GTR + F + R4 model. Bootstrap values > 75% are shown at selected nodes. The scale bar represents the average number of substitutions per site. Sequences generated in this study appear in bold. Aedes albopictus (LC431229) was included as outgroup.

**Figure 5 insects-16-01100-f005:**
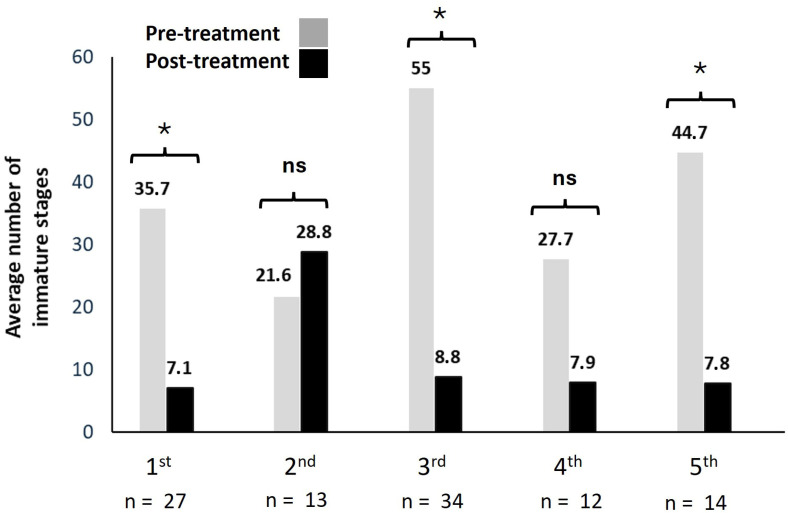
Bars represent mean larval counts per site before (grey) and after (black) each treatment round. Numbers above bars indicate mean values. Asterisks indicate significant pre–post differences (Wilcoxon signed-rank test, *p* < 0.05); ns = not significant. Sample sizes (*n*) correspond to the number of positive sites treated in each round. Effect sizes (median differences) with 95% confidence intervals are reported in the results section. Each treatment round was analyzed independently; therefore, no multiple-testing correction was applied.

## Data Availability

The original contributions presented in this study are included in the article/Appendix A. Further inquiries can be directed to the corresponding authors.

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
