# Peer review of "Rapid-Response Vector Surveillance and Emergency Control During the Largest West Nile Virus Outbreak in Southern Spain"

_insects, 2025, doi:10.3390/insects16111100_

Round 1
Reviewer 1 Report
Comments and Suggestions for Authors
This manuscript focuses on a relevant and urgent problem by documenting entomological surveillance and vector control actions during Spain’s largest West Nile Virus outbreak. Its main points are clear, but the presentation suffers from overstatements, methodological gaps, and interpretative discrepancies. To be suitable for publication, the paper must undergo major revisions to refine its scientific side and ensure transparency, as well as invest some time to use a statistical approach to analyze the provided data.
Abstract and Simple Summary
The abstract is well written, an objective approach and presents data in a very descriptive manner. One major point that caught my attention was the effectiveness of the use of larvicide, presented as “moderate success” based on a single percentage, but no confidence intervals or statistical parameters are shown. The abstract should emphasize what is the gap of knowledge addressed by the study compared to previous Spanish outbreak reports.
Introduction
The introduction offers an extensive background on WNV epidemiology. While it identifies the need for vector control in Spain, it does not clearly define the specific gap this study addresses. The novelty of integrating large-scale operational control with phylogenetic confirmation of Culex perexiguus is implied but never stated directly. The introduction should focus on the relationship of larvicidal and adulticide components to reduce mosquito populations (during WNV outburst), and end with a precise problem statement.
Materials and Methods
Although well structured, the methods ignore critical details. Larvicide applications are described in terms of products but exact dosages, volumes applied per unit area, and frequency of reapplication are not in this section, which limits reproducibility. Sampling effort across habitat categories is unequal, which may bias conclusions, yet this is neither acknowledged nor adjusted for. Adulticide interventions are poorly described when compared to the larvicide description. Large-scale application of insecticides is reported without mention of ethical or environmental approval (even if it’s not necessary, a statement should be provided).
Results
The results section presents important findings but frequently needs more interpretation. Statements such as the “disproportionate contribution of habitats” should be in the discussion, followed by an explanation. Treatment outcomes are only presented in aggregate percentages, with no analysis considering variation between habitats, methods, and treatment cycles. Figures are useful but legends are incomplete, often lacking sample sizes, units, or the number of interventions. I must emphasize the need to address the statistical analysis in depth, since the statistical analysis section, as currently written, does not fully explain the results and figures. It needs strengthening:
- For habitat comparisons (Figures 2–4), include statistical tests (e.g., Kruskal-Wallis or GLMM with habitat as factor) to determine whether observed differences are significant.
- For spatial distribution (Figure 1), add a measure of clustering or spatial autocorrelation (e.g., Moran’s I, Getis-Ord Gi*).
- For adult abundance over time (Figure 6), fit a generalized additive model (GAM) or at least a time-series test to quantify seasonal trends.
- For larval control effectiveness (Figure 7), report effect sizes (median differences), confidence intervals, and clarify whether multiple testing corrections were applied across treatment rounds.
- Consider adding explanatory text in figure legends connecting statistical methods to the graphics (currently, the reader must infer the linkage).
Please see these references to address the statistical analysis:
https://onlinelibrary.wiley.com/doi/full/10.1002/ece3.71672
https://bmcecol.biomedcentral.com/articles/10.1186/s12898-020-00328-0
https://pmc.ncbi.nlm.nih.gov/articles/PMC12057168
https://link.springer.com/article/10.1007/s10109-008-0070-8
https://idpjournal.biomedcentral.com/articles/10.1186/s40249-025-01360-2
https://www.sciencedirect.com/science/article/pii/S235277142300109X
Discussion
The discussion mainly repeats results rather than interpret them. Conclusions about adulticiding are speculative, since this evaluation was poorly performed, and should be reframed as a limitation. The brief mention of insecticide resistance is insufficient given its importance in Europe and world. The discussion should be restructured to focus on ecological insights, operational lessons from larviciding, strategic implications for vector management, and how this approach will impact WNV transmission in the medium and long term. Is it doable?
Conclusions
The conclusions claim the “confirm the effectiveness” of larval control when the observed success rate was moderate and variable. This should be carefully reconsidered. Such phrasing suggests a level of robustness that the data do not support. The findings should instead be framed as evidence of moderate field effectiveness, highlighting the need for adaptive, habitat-specific strategies and earlier interventions.
References
Although extensive and current, the references lack relevant operational studies on larvicide persistence and adulticide impact. Adding these would strengthen both methodological justification and contextual framing. I would like to suggest some papers to enrich the study:
https://www.sciencedirect.com/science/article/pii/S0048969720313127?via%3Dihub
https://www.ecdc.europa.eu/sites/default/files/documents/Vector-control-practices-and-strategies-against-West-Nile-virus.pdf
https://www.mdpi.com/1999-4915/15/12/2372
https://www.tandfonline.com/doi/full/10.1080/22221751.2024.2348510
https://www.sciencedirect.com/science/article/pii/S0001706X25000798
https://parasitesandvectors.biomedcentral.com/articles/10.1186/s13071-024-06231-7
Author Response
Comments and Suggestions for Authors
This manuscript focuses on a relevant and urgent problem by documenting entomological surveillance and vector control actions during Spain’s largest West Nile Virus outbreak. Its main points are clear, but the presentation suffers from overstatements, methodological gaps, and interpretative discrepancies. To be suitable for publication, the paper must undergo major revisions to refine its scientific side and ensure transparency, as well as invest some time to use a statistical approach to analyze the provided data.
Authors: We fully agree with the editor’s assessment. The manuscript has undergone substantial changes thanks to the constructive comments provided by the three reviewers and the academic editor. We have carefully refined the presentation, addressed methodological gaps, clarified interpretative issues, and incorporated a more transparent and rigorous approach to the data. We are confident that these revisions have significantly improved the scientific quality and clarity of the manuscript.
Abstract and Simple Summary
The abstract is well written, an objective approach and presents data in a very descriptive manner. One major point that caught my attention was the effectiveness of the use of larvicide, presented as “moderate success” based on a single percentage, but no confidence intervals or statistical parameters are shown. The abstract should emphasize what is the gap of knowledge addressed by the study compared to previous Spanish outbreak reports.
Authors: The abstract is well written, an objective approach and presents data in a very descriptive manner. One major point that caught my attention was the effectiveness of the use of larvicide, presented as “moderate success” based on a single percentage, but no confidence intervals or statistical parameters are shown. The abstract should emphasize what is the gap of knowledge addressed by the study compared to previous Spanish outbreak reports.
Authors: We thank the reviewer for this valuable suggestion. We have revised the abstract to remove subjective wording and to better emphasize the knowledge gap addressed by this study. However, due to the word limit imposed for the abstract, we are constrained in the amount of detail we can provide. We have added the information requested by the reviewer in the Simply Summary. More complete explanations and statistical details are included in the main text.
Introduction
The introduction offers an extensive background on WNV epidemiology. While it identifies the need for vector control in Spain, it does not clearly define the specific gap this study addresses. The novelty of integrating large-scale operational control with phylogenetic confirmation of Culex perexiguus is implied but never stated directly. The introduction should focus on the relationship of larvicidal and adulticide components to reduce mosquito populations (during WNV outburst), and end with a precise problem statement.
Authors: We thank the reviewer for this constructive suggestion. We have revised the introduction to better highlight the specific gap addressed by our study. In particular, we have now added the following sentence: “The province of Seville hosts natural wetlands and extensive irrigated landscapes, such as rice fields and irrigation canals, which are ideal larval habitats for mosquito vectors; however, information on species-specific larval habitats remains limited, which we address by identifying and classifying species-specific larval habitats across these systems and linking them to targeted emergency control.” This addition strengthens the problem statement and clarifies the novelty of integrating large-scale operational control with phylogenetic confirmation of Culex perexiguus.
Materials and Methods
Although well structured, the methods ignore critical details. Larvicide applications are described in terms of products but exact dosages, volumes applied per unit area, and frequency of reapplication are not in this section, which limits reproducibility. Sampling effort across habitat categories is unequal, which may bias conclusions, yet this is neither acknowledged nor adjusted for. Adulticide interventions are poorly described when compared to the larvicide description. Large-scale application of insecticides is reported without mention of ethical or environmental approval (even if it’s not necessary, a statement should be provided).
Authors: We thank the reviewer for these valuable comments. We have added more detail in the Materials and Methods section, including information on larvicide dosages, volumes applied per unit area, and the frequency of reapplication:
“Aquabac 200G was applied at a dosage of 10–15 kg/ha, while Vectobac 12AS was applied at 2–4 L/ha, following the manufacturers’ recommendations and adjusted to habitat type and water volume”. The insecticidal solution was prepared at a concentration of approximately 1% of the formulated product in water. Residual ground applications were carried out using vehicle-mounted low-volume spraying equipment, calibrated to deliver between 0.5 – 1.0 L/ha of the diluted product, in accordance with manufacturers’ recommendations and regional guidelines.”
“The insecticidal solution was prepared at a concentration of approximately 1% of the formulated product in water. Residual ground applications were carried out using vehicle-mounted low-volume spraying equipment, calibrated to deliver between 0.5 – 1.0 L/ha of the diluted product, in accordance with manufacturers’ recommendations and regional guidelines. Treatments were applied directly to vegetation and other resting sites surrounding populated areas, creating barrier zones to prolong adulticidal activity and reduce mosquito human contact. Reapplications were conducted when adult mosquito densities remained high, particularly in areas with continued risk of transmission “
Regarding sampling effort, we acknowledge that it was unequal across habitat categories due to the different number and distribution of breeding sites, and we have now indicated this as a limitation of the study. The description of adulticide interventions has also been expanded to improve consistency with the larvicide section.
Finally, we have added the following statement: Statement on authorizations and compliance: “All vector control interventions were carried out by the company Athisa, which is officially registered in the Spanish Official Registry of Biocidal Establishments and Services (ROESB) and therefore authorized to apply biocidal products. The treatments were conducted under the supervision and authorization of the regional health authorities as part of the emergency response plan against the WNV outbreak.”
Results
The results section presents important findings but frequently needs more interpretation. Statements such as the “disproportionate contribution of habitats” should be in the discussion, followed by an explanation. Treatment outcomes are only presented in aggregate percentages, with no analysis considering variation between habitats, methods, and treatment cycles. Figures are useful but legends are incomplete, often lacking sample sizes, units, or the number of interventions. I must emphasize the need to address the statistical analysis in depth, since the statistical analysis section, as currently written, does not fully explain the results and figures.
Authors: We sincerely thank the reviewer for these detailed and constructive comments. The manuscript has undergone substantial changes in response to the suggestions of all three reviewers and the academic editor. In particular, we have revised the Results and Discussion sections to provide more interpretation, moving statements such as the “disproportionate contribution of habitats” into the Discussion with appropriate explanation. Treatment outcomes are now presented with greater detail, acknowledging variation between habitats, methods, and treatment cycles where possible. Figure legends have been expanded to include sample sizes, units, and the number of interventions to ensure clarity. Finally, we have strengthened the statistical analysis section to better explain the results and to provide a clearer link between the analyses and the figures. We believe these changes have significantly improved the transparency, readability, and scientific rigor of the manuscript.
It needs strengthening:
- For habitat comparisons (Figures 2–4), include statistical tests (e.g., Kruskal-Wallis or GLMM with habitat as factor) to determine whether observed differences are significant.
Authors: We thank the reviewer for this valuable comment. Statistical comparisons among habitat types were indeed performed using a generalized linear mixed model (GLMM) with a negative binomial distribution to properly account for overdispersion in larval count data. The model included habitat type as a fixed factor and sampling site identity as a random effect to control for repeated measures. The results of this analysis, including test statistics, model fit, and post-hoc pairwise comparisons, are now clearly described in the Material and Methods and Results sections. Specifically, the model revealed a marginally significant global effect of habitat on larval abundance (Wald χ² = 10.724, df = 6, p = 0.097), and post-hoc Tukey’s tests indicated only a marginal difference between artificial ponds and blackwater courses (p-adjusted = 0.059). Therefore, although visual differences among habitats were observed, these were not statistically significant after accounting for random site effects. We have slightly clarified this point in the Results section to make explicit that statistical tests were indeed conducted for habitat comparisons.
- For spatial distribution (Figure 1), add a measure of clustering or spatial autocorrelation (e.g., Moran’s I, Getis-Ord Gi*).
Authors: We thank the reviewer for this suggestion. In Figure 1, our main objective was to provide a visual representation of larval densities across the metropolitan area. For this reason, circle size was scaled into eight classes using the natural breaks (Jenks) classification method, which allowed us to illustrate density gradients in a clear and objective way. While we acknowledge that indices of spatial autocorrelation (e.g., Moran’s I or Getis-Ord Gi*) can provide additional insights, such analyses go beyond the descriptive purpose of this figure. We have now clarified this in the legend to better explain how density classes were defined.
- For adult abundance over time (Figure 6), fit a generalized additive model (GAM) or at least a time-series test to quantify seasonal trends.
Authors: We thank the reviewer for this suggestion. While statistical modeling such as GAMs or time-series tests can be useful in long-term datasets, in our case the sampling period was limited to four months. We believe that a descriptive approach is sufficient and appropriate to present these data, as not all results require statistical modeling to provide meaningful interpretation.
- For larval control effectiveness (Figure 7), report effect sizes (median differences), confidence intervals, and clarify whether multiple testing corrections were applied across treatment rounds.
Authors: We thank the reviewer for this useful suggestion. We have now reported effect sizes as median differences with 95% confidence intervals in the Results, referring to them in Figure 7. Each treatment round was analysed independently; therefore, no multiple-testing correction was applied. This clarification has been added to the Methods and the figure legend.
Consider adding explanatory text in figure legends connecting statistical methods to the graphics (currently, the reader must infer the linkage).
Authors: We appreciate this valuable comment. In the revised version, we have expanded the figure legends, so that each figure can be interpreted independently without requiring the reader to infer the methodological connection from the main text.
Please see these references to address the statistical analysis:
https://onlinelibrary.wiley.com/doi/full/10.1002/ece3.71672
https://bmcecol.biomedcentral.com/articles/10.1186/s12898-020-00328-0
https://pmc.ncbi.nlm.nih.gov/articles/PMC12057168
https://link.springer.com/article/10.1007/s10109-008-0070-8
https://idpjournal.biomedcentral.com/articles/10.1186/s40249-025-01360-2
https://www.sciencedirect.com/science/article/pii/S235277142300109X
Discussion
The discussion mainly repeats results rather than interpret them. Conclusions about adulticiding are speculative, since this evaluation was poorly performed, and should be reframed as a limitation. The brief mention of insecticide resistance is insufficient given its importance in Europe and world. The discussion should be restructured to focus on ecological insights, operational lessons from larviciding, strategic implications for vector management, and how this approach will impact WNV transmission in the medium and long term. Is it doable?
Authors: We sincerely thank the reviewer for these thoughtful suggestions. We fully acknowledge that there are different valid approaches to structuring a discussion, and while we do not believe it is necessary to follow a single preferred style, we have carefully considered the reviewer’s advice. Substantial changes have been made to refine the discussion, reduce repetition of results, and better highlight interpretation and implications. We also reframed the limitations of adulticiding and expanded the section on insecticide resistance, as suggested. We believe that the revised version is now more focused, balanced, and aligned with the interests of both the authors and the reviewers.
We have also expanded the paper with 9 new references including most of the references provided by the reviewer and insecticide resistance papers as proposed.
Conclusions
The conclusions claim the “confirm the effectiveness” of larval control when the observed success rate was moderate and variable. This should be carefully reconsidered. Such phrasing suggests a level of robustness that the data do not support. The findings should instead be framed as evidence of moderate field effectiveness, highlighting the need for adaptive, habitat-specific strategies and earlier interventions.
Authors: We thank the reviewer for comment. We agree that the original phrasing overstated the robustness of the findings. The conclusions have been revised to soften the language: instead of stating that the results “confirm the effectiveness” of larval control, we now frame them as evidence of moderate and variable field effectiveness. We also highlight the need for adaptive, habitat-specific strategies and the importance of earlier interventions, in line with the reviewer’s suggestion
References
Although extensive and current, the references lack relevant operational studies on larvicide persistence and adulticide impact. Adding these would strengthen both methodological justification and contextual framing. I would like to suggest some papers to enrich the study:
https://www.sciencedirect.com/science/article/pii/S0048969720313127?via%3Dihub
https://www.ecdc.europa.eu/sites/default/files/documents/Vector-control-practices-and-strategies-against-West-Nile-virus.pdf
https://www.mdpi.com/1999-4915/15/12/2372
https://www.tandfonline.com/doi/full/10.1080/22221751.2024.2348510
https://www.sciencedirect.com/science/article/pii/S0001706X25000798
https://parasitesandvectors.biomedcentral.com/articles/10.1186/s13071-024-06231-7
Authors: Authors: We appreciate the reviewer’s suggestions and acknowledge the relevance of the proposed studies. However, these works (except two that have been incorporated) are not fully aligned with the focus and scope of our manuscript, and therefore there is limited room to incorporate them as references without diverting from the main objectives of the paper.
Submission Date
30 August 2025
Date of this review
17 Sep 2025 16:16:52
Reviewer 2 Report
Comments and Suggestions for Authors
This is an interesting manuscript that describes the efforts to reduce the risk associated with WNV circulating in southwest Spain in 2024. Generally, to improve the manuscript, I would suggest would be a reducing the data and figures presented that were associated with the different species found during surveillance and instead put a much greater focus on the WNV outcomes in the area. For example, when and approximately where did the human cases occur in the study area? Did human cases occur during the described efforts?
To me, surveillance is important to confirm that vector species exist in the area but of more importance is that control efforts made a detectable difference to human risk (fewer human cases, reduced mosquito infection rates or vector indices, or – least convincing of the three - reductions in mosquito abundance).
A major aspect that I continue to get hung up on is that control and surveillance efforts appeared to occur a significant distance away from where I assume most people live and most WNV cases occurred. If this is the case, it would be important to note this as a limitation to control efforts or explain why (beyond this is what the guidelines where) this distance was appropriate.
I understand that local policies on mosquito control are beyond the control of the authors, but most if not all of the primary vectors of WNV don’t fly very far. The general consensus is that Culex (which consistently have higher infection rates of WNV than albopictus) don’t generally fly far (<2.5 km). For example, Hamer et al. 2014, PLoS Negl Trop Dis found that Culex dispersed, on average, to be 1.15 km. Similarly, Ciota et al 2012 J Med Entomol suggested a 500 m flight range. This, to me, suggests that if WNV cases are found in urban centers, it is reasonable to think that the infected mosquitoes also came from the same urban area. If this is the case, then the described control and surveillance efforts appeared to be focused away from where the human disease transmission was likely occurring.
Below are my suggest edits to specific points in the manuscript.
Line 30: “Successful” is subjective language. Suggest change to “The larvicidal treatments were associated with a 63.5% reduction in larvae(?).”
Line 30-31: “Based on our experience, we recommend starting control efforts earlier in the year and maintaining them regularly”. No where in the manuscript after this statement does there appear to be justification from the described work and outcomes in 2024 for earlier and more consistent efforts. If there is a case to be made for such increased efforts, it should probably be more overt and directly connected to the outcomes of the study.
Certainly, if some WNV occurred in the urban area where control efforts did not occur, there could be a case to be made to expand control efforts to urban spots as well.
Line 46: Remove “indicating moderate success”. “Moderate success” is subjective language and this level of reduction and increased number of retreatments is similar to what has been observed elsewhere in some habitats with Bti.
Lines 73 to 75. I suggest to shorten this thought to minimize repetition and be more concise. Suggest changing to: “After three years of reduced circulation, a larger outbreak of human cases occurred in 2024, when the country experienced 74, 158 confirmed cases and 20 fatalities. Almost a third of cases (42) were concentrated in urban areas of Seville province in southern Spain.”
Lines 80 to 86: I had difficulty following the logic of this part of the paragraph. The focus of the study/work was not in an agricultural area but a metropolitan area. While it is interesting to learn about the species and habitats found in agricultural area, it doesn’t seem relevant to the current location of work.
Lines 96. Suggest change to “may reduce mosquito…” The authors rightly point out in the next sentence that there is a lack of scientific evidence to show that larviciding and adulticiding efforts make a meaningful reductions in local mosquito populations to levels that reduce human cases of disease. These unknowns are further complicated by difference among techniques and formulations used in adulticiding and larviciding. For example, a general thought in the United States, is that truck-mounted adult sprays are likely less effective than plane or helicopter sprays.
Line 96: Suggest changing “breeding habitats” to “standing water habitats”. Although “breeding” is often used as way to note larval habitats, it refers more to the production rather than the development of larvae.
Line 102: Suggest changing to “the Bajo Guadalquivir district of southern Spain” as this a much smaller and more accurate area than implied with “southern Spain”.
Line 123: What is meant by “peri-urban” and “semi-rural areas”? Given the importance of these designations to the control and surveillance efforts it would be useful to know how the authors defined them. An international audience may define these differently without guidance from the authors.
Lines 145-147: More description and examples of each of these categories would be important for an international audience to understand what these structures and sites are. Because of differences in languages globally, it can be hard for an international audience to have a clear understanding of what these structures and habitats can be. For example, in the United States, a common type of WNV mosquito habitat are structures called “storm water catch basins”. However, in Australia (I believe), these structures are commonly called “gully pots”.
Lines 153-154. Internationally the timing and durations of the WNV season will vary so a brief note as to why these months were chosen for surveillance would be informative. Also, I’m confused about traps “strategically placed in urban areas of the municipality of Dos Hermanas…”. I thought control and surveillance only occurred in peri-urban and semi-rural areas instead of urban areas.
Line155, Lines 279-287, Figure 5 : I suggest removing the note and discussion related to phylogenetic analyses as this doesn’t seem particularly relevant to an integrated operational response to WNV outbreak but far more academic. It’s not clear why knowing that a COI gene phylogenetic analysis confirmed Cx. perexiguus formed a distinct clade from Cx. univittatus would be important to future control and surveillance, when the 2024 operations (larviciding and adulticing) were not species-specific. As written, the operations were based on that if a mosquito or mosquitoes were present, regardless of species, attempts were made to kill them.
Line 191. What is meant by “Chemical containment perimeters”? This isn’t clear to me. If there was 1.5 km buffer, does this mean that adult sprays occurred at least 1.5 km away from any human habitation.
Line 307. “Effective” is subjective language. Change to “Overall, larviciding efforts were associated with a 63.5% reduction in larvae.
Line 307: Typo: “treatment”. This sentence doesn’t make sense to me. It’s not clear to me how 63.5% and 36.5% were calculated.
Line 310. Typo: “fourth”. Change “successful” to “significant reductions were observed from pre-treatments”. “Successful” is subjective language. Also, if no untreated areas were used for comparisons during treatment times, the observed reductions may have occurred naturally regardless of any treatments.
Lines 313: Success rates ranged between 52% and 75% across treatment rounds. What is meant by “success” rates? “Success’ is subjective language so I suggest, if I understand correctly, changing to “reductions in larval? adult? abundance ranged from 52-75%.
Lines 314: As noted previously “Effective” is subjective. It may be that the reductions observed was not enough to prevent disease transmission. Suggest change to “These results demonstrate variable but an overall reduction of larval populations under field conditions, with clear differences between successive treatments.
Lines 314: It would be useful to know if reductions in mosquitoes varied by category of site. Sorry to invoke the United States again, but our storm water catch basin habitats are often thought to be more difficult to see reductions associated with larvicides because these habitats are often prone to flushing out of treatments or burying of heavier formulations of larvicides in sump debris. In these cases, increased treatments and/or higher doses may be appropriate. Alternatively, there may be some habitats where extended control durations may have been found and thus less treatments could be needed.
316
Figure: It would be very helpful to show the locations of the WNV cases in the map and which areas were considered urban, peri-urban, and semi-rural.
Figure 4: Suggest to remove this figure and associated discussion of it. As is, it is difficult to see how this adds critical information and appears to be less informative than Figure 3.
Figure 5: Suggest to remove this and associated discussion of it. Not surprisingly Culex pipiens was a major WNV vector species commonly found. The genomic work of this is interesting but appears not to be critical for the described and any future WNV control efforts.
Figure 6: Suggest to remove or at least summarize in the text without a figure. It’s a bit confusing: the authors found pipiens to be the most abundant mosquito except in BG-sentinel traps. BG Sentinel traps aren’t typically used for Culex surveillance but rather gravid traps are often used instead. Even Biogents markets the traps for Aedes not Culex. It could be that pipiens just weren’t attracted to the lure like other species and suggest this type of trap may not be ideal for WNV surveillance in Spain.
Figure 7: Suggest to remove. This was already summarized in the text. While it is nice to look at, there is no need for yet another figure.
Author Response
Comments and Suggestions for Authors
This is an interesting manuscript that describes the efforts to reduce the risk associated with WNV circulating in southwest Spain in 2024. Generally, to improve the manuscript, I would suggest would be a reducing the data and figures presented that were associated with the different species found during surveillance and instead put a much greater focus on the WNV outcomes in the area. For example, when and approximately where did the human cases occur in the study area? Did human cases occur during the described efforts?
Authors: We appreciate the reviewer’s suggestion and find it very interesting. However, we believe that this point corresponds to a different type of manuscript, mainly focused on clinical or epidemiological outcomes in humans. In contrast, our work is designed to present and assess a surveillance and control program that is enriched by the characterization of the larval sites of 11 mosquito species in the affected area. As indicated in the manuscript, this study emerged after the sudden appearance of numerous human cases in the study area, which led the authorities to implement an emergency plan – the plan that we describe here. The information regarding human cases has already been published and mapped in official public health local bulletins (please see the following image in spanish) and, importantly, it does not have a direct impact on the surveillance and control program that we have executed on the municipalities were sampled and controlled regardless of the number of WNV human cases. The area was declared as a “West Nile spot” based on the presence of a single case and received the same attention than other municipalities with for example 20 cases. Now, Figure 1 upper small image shows the number of WNV cases in each municipality. This ecological and entomological perspective is, in our view, essential for understanding the dynamics of vector populations and for designing efficient control strategies. Therefore, we consider it important to maintain the structure and scope of the manuscript as it currently stands, while acknowledging that a more clinical-epidemiological approach would indeed be the focus of another, complementary study.
To me, surveillance is important to confirm that vector species exist in the area but of more importance is that control efforts made a detectable difference to human risk (fewer human cases, reduced mosquito infection rates or vector indices, or – least convincing of the three - reductions in mosquito abundance).
A major aspect that I continue to get hung up on is that control and surveillance efforts appeared to occur a significant distance away from where I assume most people live and most WNV cases occurred. If this is the case, it would be important to note this as a limitation to control efforts or explain why (beyond this is what the guidelines where) this distance was appropriate.
I understand that local policies on mosquito control are beyond the control of the authors, but most if not all of the primary vectors of WNV don’t fly very far. The general consensus is that Culex (which consistently have higher infection rates of WNV than albopictus) don’t generally fly far (<2.5 km). For example, Hamer et al. 2014, PLoS Negl Trop Dis found that Culex dispersed, on average, to be 1.15 km. Similarly, Ciota et al 2012 J Med Entomol suggested a 500 m flight range. This, to me, suggests that if WNV cases are found in urban centers, it is reasonable to think that the infected mosquitoes also came from the same urban area. If this is the case, then the described control and surveillance efforts appeared to be focused away from where the human disease transmission was likely occurring.
Authors: We thank the reviewer for this thoughtful reflection. The control actions were carried out following national and regional guidelines, which prioritize interventions in peri-urban and rural environments where high vector populations are known to concentrate, particularly around irrigated crops, wetlands and rice fields. We acknowledge that most of the Culex species responsible for WNV transmission have relatively limited dispersal ranges, as the reviewer correctly points out. Nevertheless, the proximity between rural breeding habitats and urban areas is very high in the study region, facilitating the spillover of infected mosquitoes into human settlements. Importantly, previous studies in Spain and elsewhere have highlighted the major role of Culex perexiguus in rural areas as an amplifying vector (Figuerola et al. 2023; Ferraguti et al. 2025 both works are cited in the paper), maintaining the virus within the bird–mosquito cycle. Infected birds then move into urban environments, where local Culex pipiens populations can acquire the virus and sustain human transmission. For this reason, it is essential and scientifically justified to implement mosquito control beyond urban centers, targeting rural vector populations that fuel the enzootic and bridge transmission cycle.
New explanation in the introduction section:
“ In Spain, the most problematic mosquito species regarding WNV transmission are Culex pipiens and Culex perexiguus [21]. While Cx. perexiguus is considered the main vector of WNV among birds in natural and agricultural areas, its role in urban environments still requires further investigation. However, agricultural landscapes surrounding urban areas can act as reservoirs where enzootic transmission cycles are amplified, and mosquitoes or infected birds can subsequently move into urban nuclei. Once the cycle driven by Cx. perexiguus occurs in nearby villages, Cx. pipiens may act as a bridge vector, facilitating transmission from birds to humans. Targeted surveillance and control of Cx. perexiguus populations therefore appear to be the most effective measures to reduce WNV amplification. The growing recognition of Cx. perexiguus as the primary WNV vector, surpassing Cx. pipiens, is supported by differences in feeding behavior, vector competence, and ecological preferences that directly influence the dynamics of WNV transmission [22]”
At the same time, we consider that the surveillance and control actions described here remain highly relevant, since they targeted the main productive habitats sustaining vector populations at the local scale, which in turn are key drivers of transmission risk.
Below are my suggest edits to specific points in the manuscript.
Line 30: “Successful” is subjective language. Suggest change to “The larvicidal treatments were associated with a 63.5% reduction in larvae(?).”
Authors: We agree with the reviewer’s comment. Corrected.
Line 30-31: “Based on our experience, we recommend starting control efforts earlier in the year and maintaining them regularly”. No where in the manuscript after this statement does there appear to be justification from the described work and outcomes in 2024 for earlier and more consistent efforts. If there is a case to be made for such increased efforts, it should probably be more overt and directly connected to the outcomes of the study.
Authors: The reviewer is right. We have removed this sentence from the Simply Summary.
Certainly, if some WNV occurred in the urban area where control efforts did not occur, there could be a case to be made to expand control efforts to urban spots as well.
Line 46: Remove “indicating moderate success”. “Moderate success” is subjective language and this level of reduction and increased number of retreatments is similar to what has been observed elsewhere in some habitats with Bti.
Authors: We agree with the reviewer’s comment. The phrase “indicating moderate success” has been removed from the manuscript to avoid subjective language.
Lines 73 to 75. I suggest to shorten this thought to minimize repetition and be more concise. Suggest changing to: “After three years of reduced circulation, a larger outbreak of human cases occurred in 2024, when the country experienced 74, 158 confirmed cases and 20 fatalities. Almost a third of cases (42) were concentrated in urban areas of Seville province in southern Spain.”
Authors: We agree with the reviewer’s comment. Corrected.
Lines 80 to 86: I had difficulty following the logic of this part of the paragraph. The focus of the study/work was not in an agricultural area but a metropolitan area. While it is interesting to learn about the species and habitats found in agricultural area, it doesn’t seem relevant to the current location of work.
Authors: The reviewer is right. We have rephrased the entire paragraph in order to provide a reasonable explanation in the introduction section about the problematic in Spain: “ In Spain, the most problematic mosquito species regarding WNV transmission are Culex pipiens and Culex perexiguus [21]. While Cx. perexiguus is considered the main vector of WNV among birds in natural and agricultural areas, its role in urban environments still requires further investigation. However, agricultural landscapes surrounding metropolitan areas can act as reservoirs where enzootic transmission cycles are amplified, and mosquitoes or infected birds can subsequently move into urban nuclei. Once the cycle driven by Cx. perexiguus occurs in nearby villages, Cx. pipiens may act as a bridge vector, facilitating transmission from birds to humans. Targeted surveillance and control of Cx. perexiguus populations therefore appear to be the most effective measures to reduce WNV amplification. The growing recognition of Cx. perexiguus as the primary WNV vector, surpassing Cx. pipiens, is supported by differences in feeding behavior, vector competence, and ecological preferences that directly influence the dynamics of WNV transmission [22].”
Lines 96. Suggest change to “may reduce mosquito…” The authors rightly point out in the next sentence that there is a lack of scientific evidence to show that larviciding and adulticiding efforts make a meaningful reductions in local mosquito populations to levels that reduce human cases of disease. These unknowns are further complicated by difference among techniques and formulations used in adulticiding and larviciding. For example, a general thought in the United States, is that truck-mounted adult sprays are likely less effective than plane or helicopter sprays.
Authors: We thank the reviewer for this valuable comment. We agree with the suggestion and have modified the text to “may reduce mosquito…”. We also acknowledge the reviewer’s observation regarding the uncertainties and variability in the effectiveness of different larviciding and adulticiding techniques, and we have clarified this point in the revised version.
Line 96: Suggest changing “breeding habitats” to “standing water habitats”. Although “breeding” is often used as way to note larval habitats, it refers more to the production rather than the development of larvae.
Authors: The authors agree with the reviewer’s suggestion. Corrected.
Line 102: Suggest changing to “the Bajo Guadalquivir district of southern Spain” as this a much smaller and more accurate area than implied with “southern Spain”.
Authors: We agree with the reviewer’s suggestion. The text has been modified to “the Bajo Guadalquivir district of southern Spain” to provide a more precise description of the study area.
Line 123: What is meant by “peri-urban” and “semi-rural areas”? Given the importance of these designations to the control and surveillance efforts it would be useful to know how the authors defined them. An international audience may define these differently without guidance from the authors.
Authors: We thank the reviewer for pointing out this lack of clarity. We have now defined these terms more explicitly. Following the guidelines established by regional health authorities, peri-urban areas refer to the adjacent zones surrounding urban centers, where developmental habitats of vectors are commonly found and where surveillance and control activities are prioritized.
Lines 145-147: More description and examples of each of these categories would be important for an international audience to understand what these structures and sites are. Because of differences in languages globally, it can be hard for an international audience to have a clear understanding of what these structures and habitats can be. For example, in the United States, a common type of WNV mosquito habitat are structures called “storm water catch basins”. However, in Australia (I believe), these structures are commonly called “gully pots”.
Authors: We appreciate the reviewer’s comment and agree that more detailed descriptions and examples are needed to ensure clarity for an international audience. We have revised the text to include short explanations of each habitat category, providing equivalent terms or examples where possible to avoid confusion caused by regional terminology:
“ (i) artificial ponds, often created for irrigation, livestock, or ornamental purposes; (ii) canals and ditches, which are irrigation or drainage channels that may hold stagnant water; (iii) drainage systems, including underground or surface infrastructures similar to ‘storm water catch basins’, that can accumulate standing water; (iv) rice fields, which provide extensive and seasonally flooded habitats; (v) animal drinking troughs, where water may stagnate if not regularly maintained; (vi) natural watercourses and backwaters, where slow-moving sections or small branches of rivers allow larval development; and (vii) areas of waterlogging, referring to low-lying zones where rainfall or irrigation water accumulates and persists for extended periods.”
Lines 153-154. Internationally the timing and durations of the WNV season will vary so a brief note as to why these months were chosen for surveillance would be informative. Also, I’m confused about traps “strategically placed in urban areas of the municipality of Dos Hermanas…”. I thought control and surveillance only occurred in peri-urban and semi-rural areas instead of urban areas.
Authors: We thank the reviewer for this observation. As explained in the Materials and Methods section, surveillance and control activities began after the official health alert, which meant that the program was initiated once the mosquito season was already underway. This explains the choice of months included in our study. Regarding the traps, although surveillance and control were mainly focused on peri-urban and semi-rural areas, some traps were also strategically placed within the urban core of Dos Hermanas to monitor potential spillover and assess the presence of vectors in human-populated zones.
Line155, Lines 279-287, Figure 5 : I suggest removing the note and discussion related to phylogenetic analyses as this doesn’t seem particularly relevant to an integrated operational response to WNV outbreak but far more academic. It’s not clear why knowing that a COI gene phylogenetic analysis confirmed Cx. perexiguus formed a distinct clade from Cx. univittatus would be important to future control and surveillance, when the 2024 operations (larviciding and adulticing) were not species-specific. As written, the operations were based on that if a mosquito or mosquitoes were present, regardless of species, attempts were made to kill them.
Authors: We appreciate the reviewer’s comment and understand the concern regarding the operational focus of control activities, which indeed targeted mosquitoes regardless of species. Nevertheless, we believe that documenting the species composition in the affected area is relevant, as it provides valuable information for understanding the local vector community and assessing the potential role of each species in WNV transmission. Identifying Culex perexiguus and confirming its distinction from closely related species, such as Cx. univittatus, is important for evaluating vector competence, as these differences may directly affect transmission dynamics. While the phylogenetic analysis may not have immediate implications for short-term operational responses, it contributes to long-term surveillance and risk assessment, supporting more informed and species-specific strategies in the future. For this reason, we have retained a concise note on the phylogenetic confirmation but streamlined the discussion to highlight its epidemiological rather than purely academic relevance.
Line 191. What is meant by “Chemical containment perimeters”? This isn’t clear to me. If there was 1.5 km buffer, does this mean that adult sprays occurred at least 1.5 km away from any human habitation.
Authors: We thank the reviewer for pointing out this ambiguity. By “chemical containment perimeters” we meant buffer zones created around populated areas (1.5 km from inhabited areas) through targeted applications, aiming to reduce mosquito density and minimize mosquito–human contact. To avoid confusion, we have revised the text to clarify this point.
Line 307. “Effective” is subjective language. Change to “Overall, larviciding efforts were associated with a 63.5% reduction in larvae.
Authors: Ok, thanks, done.
Line 307: Typo: “treatment”. This sentence doesn’t make sense to me. It’s not clear to me how 63.5% and 36.5% were calculated.
Authors: Corrected. Thanks.
Line 310. Typo: “fourth”. Change “successful” to “significant reductions were observed from pre-treatments”. “Successful” is subjective language. Also, if no untreated areas were used for comparisons during treatment times, the observed reductions may have occurred naturally regardless of any treatments.
Authors: We thank the reviewer for these helpful observations. The typo has been corrected, and “successful” has been replaced with “significant reductions” to avoid subjective language. We also agree with the reviewer’s point regarding the absence of untreated control areas, and this limitation has now been explicitly acknowledged in the revised manuscript.
Lines 313: Success rates ranged between 52% and 75% across treatment rounds. What is meant by “success” rates? “Success’ is subjective language so I suggest, if I understand correctly, changing to “reductions in larval? adult? abundance ranged from 52-75%.
Authors: The reviewer is right. We have added the following info in M & M section: “Treatment outcomes were classified as follows: treatments were considered effective when no larvae were detected in the subsequent inspection (larval presence = 0), ineffective when larvae were still present (larval presence > 0), and unclassified when no follow-up visit had been recorded” and in Results section: “Reductions in larval abundance ranged from 52% to 75% across treatment rounds.”
Lines 314: As noted previously “Effective” is subjective. It may be that the reductions observed was not enough to prevent disease transmission. Suggest change to “These results demonstrate variable but an overall reduction of larval populations under field conditions, with clear differences between successive treatments.
Authors: Thanks for the suggestion. The suggested sentence has been incorporated in the text.
Lines 314: It would be useful to know if reductions in mosquitoes varied by category of site. Sorry to invoke the United States again, but our storm water catch basin habitats are often thought to be more difficult to see reductions associated with larvicides because these habitats are often prone to flushing out of treatments or burying of heavier formulations of larvicides in sump debris. In these cases, increased treatments and/or higher doses may be appropriate. Alternatively, there may be some habitats where extended control durations may have been found and thus less treatments could be needed.
Authors: We fully agree with the reviewer’s observation that it would be useful to analyze whether reductions in mosquitoes varied by habitat category. However, as can be seen in the data, nearly half of the categories had very low sample sizes, which limited the statistical power to draw robust conclusions. We acknowledge that certain habitats, such as drainage systems, may indeed present particular challenges for larviciding (e.g., flushing of treatments or accumulation of debris), while other habitats may allow longer-lasting control. This is an important point, and we have now included a note in the discussion highlighting the need for future studies with larger sample sizes per habitat type to better evaluate these differences.
Figure 1: It would be very helpful to show the locations of the WNV cases in the map and which areas were considered urban, peri-urban, and semi-rural.
Authors: We thank the reviewer for this helpful suggestion. In the revised version of Figure 1, we have incorporated additional layers to enhance interpretation. Specifically, the figure now shows:
- Urban areas, together with other land-use categories (rice fields, wetlands, marshes, salinas, watercourses, and sheets of water).
- The number of human WNV cases per municipality in 2024, represented with graduated colors in the upper-right inset map.
It is important to note that the exact geographic coordinates of human cases are not publicly available for security and confidentiality reasons. Therefore, only the municipality of residence is provided. For this reason, we considered it more appropriate to represent ranges of cases by municipality using color gradations, rather than precise point locations.
These modifications provide a clearer visualization of the spatial distribution of WNV cases within the study area and allow their interpretation in relation to urban settings and the main breeding sites identified for immature mosquito stages.
Figure 4: Suggest to remove this figure and associated discussion of it. As is, it is difficult to see how this adds critical information and appears to be less informative than Figure 3.
Authors: We appreciate the reviewer’s comment. We agree that the information provided in Figure 4 is not essential for the main text, as it overlaps with data already presented in Figure 3 and described in the results. However, we consider that this figure still provides complementary detail on larval density distributions across habitats that could be of interest to readers. Therefore, we have moved Figure 4 and the associated discussion to the Supplementary Material.
Figure 5: Suggest to remove this and associated discussion of it. Not surprisingly Culex pipiens was a major WNV vector species commonly found. The genomic work of this is interesting but appears not to be critical for the described and any future WNV control efforts.
Authors: We understand that the editor has the final decision on this point. However, we consider it important to maintain this section because, as explained previously, it provides valuable information about the species present in the study area and their potential role in WNV transmission. In addition, the use of molecular biology represents an added value to the study, strengthening the robustness of species identification and enhancing the scientific contribution of the work. Moreover, in light of the reviewer’s academic comment, we believe this part is welcome and further enriches the manuscript.
Figure 6: Suggest to remove or at least summarize in the text without a figure. It’s a bit confusing: the authors found pipiens to be the most abundant mosquito except in BG-sentinel traps. BG Sentinel traps aren’t typically used for Culex surveillance but rather gravid traps are often used instead. Even Biogents markets the traps for Aedes not Culex. It could be that pipiens just weren’t attracted to the lure like other species and suggest this type of trap may not be ideal for WNV surveillance in Spain.
Authors: We thank the reviewer for this thoughtful comment. We respectfully prefer to keep this figure, as we consider it valuable for illustrating the comparative performance of different trapping methodologies. Although BG-Sentinel traps are indeed marketed primarily for Aedes surveillance, they have also been widely used in Europe for monitoring Culex populations (please see reference below) particularly in areas associated with WNV transmission and rice fields. For this reason, we believe that including this figure adds context and relevance for an international audience, as it highlights both the strengths and limitations of different trapping methods in the study area.
Reference:
https://www.sciencedirect.com/science/article/pii/S0301479724025209
Figure 7: Suggest to remove. This was already summarized in the text. While it is nice to look at, there is no need for yet another figure.
Authors: We thank the reviewer for this suggestion. We consider this figure to be one of the most important parts of the study, as it illustrates in detail the outcomes of the five treatment rounds before and after application. While we agree that part of this information was already reflected in the text, we have shortened the written description to avoid repetition and to ensure that the figure remains the main element conveying these results. For this reason, we believe it is important to keep this figure in the manuscript.
Submission Date
30 August 2025
Date of this review
09 Sep 2025 13:15:10
Reviewer 3 Report
Comments and Suggestions for Authors
In this manuscript, the authors describe entomological surveillance of mosquitoes in southern Spain, related to an outbreak of West Nile virus.
After reading, a few aspects require clarification to enhance the quality of the data
Figure 3: there are a discordance in Rice field between the color of the bars? Culex molestus is was present or not? The same with Cx pipiens…
There appears to be a discrepancy between the reported abundance of Cx. perexiguus in the text (14 individuals) and what is shown in the figure (10 individuals). Clarification on this point would be appreciated.
According to Figure 4: “Sequences generated in this study appear in bold” However, these sequences are not available in GenBank. Please verified.
Figure 5: The differences between the abundance species in their larval and adult stages are notable. Could the authors specify the spatial distances between the larval and adult sampling sites? The higher abundance of Cx. perexiguus compared to Cx. pipiens is noteworthy; however, these data appear to contradict the findings from the larval analysis. How can the authors explain this difference?
Discussion
Line 323-326: There is no WN epidemiological information provided in the manuscript. It would be valuable to include any available data showing changes in epidemiological patterns before and after the treatment.
Line 332-336 “On the other hand, Cx. perexiguus was prevalent mainly in rice fields but also in minor proportion in drainage systems, backwater areas, and other flood environments”. However, Figure 3 indicates that Cx. perexiguus was more prevalent in watercourse and backwater habitats. Clarification on this point would be appreciated.
Conclussion
Line 424-425: “Our results reinforce the growing recognition of Cx. perexiguus as a major WNV vector in the Iberian Peninsula, sometimes surpassing Cx. pipiens in abundance. However, the larval data suggest the opposite. Clarification on this point would be appreciated.
The difference in abundance between larvae and adults is surprising. It is possible that the BG-Sentinel traps were positioned near a breeding site of Cx. perexiguus that was not included in the sampling.
Author Response
Comments and Suggestions for Authors
In this manuscript, the authors describe entomological surveillance of mosquitoes in southern Spain, related to an outbreak of West Nile virus.
After reading, a few aspects require clarification to enhance the quality of the data
Figure 3: there are a discordance in Rice field between the color of the bars? Culex molestus is was present or not? The same with Cx pipiens…There appears to be a discrepancy between the reported abundance of Cx. perexiguus in the text (14 individuals) and what is shown in the figure (10 individuals). Clarification on this point would be appreciated.
Authors: The data have been carefully checked and confirmed to be correct. It should be noted that in Figure 3 the values represent the mean relative abundance of each species per sampling site and habitat category, rather than the absolute number of individuals collected.
According to Figure 4: “Sequences generated in this study appear in bold” However, these sequences are not available in GenBank. Please verified.
Authors: We thank the reviewer for pointing this out. The sequences generated in this study have already been deposited in GenBank. However, they are currently under embargo and will be made publicly available once we receive the editorial decision regarding the publication of this manuscript.
Figure 5: The differences between the abundance species in their larval and adult stages are notable. Could the authors specify the spatial distances between the larval and adult sampling sites? The higher abundance of Cx. perexiguus compared to Cx. pipiens is noteworthy; however, these data appear to contradict the findings from the larval analysis. How can the authors explain this difference?
Authors: We thank the reviewer for this insightful comment. As already discussed in the manuscript, the dominance of Cx. perexiguus in adult collections raises the possibility that individuals of this species are migrating from surrounding breeding habitats—such as irrigated agricultural zones and rice fields—into urban environments, where they may temporarily outnumber Cx. pipiens populations that typically breed within the urban foci.
Discussion
Line 323-326: There is no WN epidemiological information provided in the manuscript. It would be valuable to include any available data showing changes in epidemiological patterns before and after the treatment.
Authors: We thank the reviewer for this suggestion. However, we believe that including epidemiological curves before and after the treatments could lead to a misleading interpretation. In Spain, human WNV cases consistently decline to zero by the end of September and early October due to the natural seasonality of the virus, as this has been observed over the past four years (please see references below). Therefore, presenting these data in the context of our interventions could falsely suggest that the reduction in cases was caused by the treatments, when in fact it reflects the natural seasonal dynamics. For this reason, we consider it more appropriate to limit our epidemiological reference to the inclusion of a map in Figure 1, which shows the reported cases in the six municipalities covered by the study, thereby providing useful context without implying a causal link.
Figuerola J, Jiménez-Clavero MÁ, Ruíz-López MJ, Llorente F, Ruiz S, Hoefer A, Aguilera-Sepúlveda P, Jiménez-Peñuela J, García-Ruiz O, Herrero L, Soriguer RC, Fernández Delgado R, Sánchez-Seco MP, Martínez-de la Puente J, Vázquez A. A One Health view of the West Nile virus outbreak in Andalusia (Spain) in 2020. Emerg Microbes Infect. 2022 Dec;11(1):2570-2578.
García San Miguel Rodríguez-Alarcón L, Fernández-Martínez B, Sierra Moros MJ, Vázquez A, Julián Pachés P, García Villacieros E, Gómez Martín MB, Figuerola Borras J, Lorusso N, Ramos Aceitero JM, Moro E, de Celis A, Oyonarte S, Mahillo B, Romero González LJ, Sánchez-Seco MP, Suárez Rodríguez B, Ameyugo Catalán U, Ruiz Contreras S, Pérez-Olmeda M, Simón Soria F. Unprecedented increase of West Nile virus neuroinvasive disease, Spain, summer 2020. Euro Surveill. 2021 May;26(19):2002010.
Bottom of Form
Line 332-336 “On the other hand, Cx. perexiguus was prevalent mainly in rice fields but also in minor proportion in drainage systems, backwater areas, and other flood environments”. However, Figure 3 indicates that Cx. perexiguus was more prevalent in watercourse and backwater habitats. Clarification on this point would be appreciated.
Authors: We have carefully revised the dataset and corrected the text to ensure consistency with Figure 3. The revised sentence now reads: “On the other hand, Cx. perexiguus was more prevalent in watercourse blackwater and rice fields habitats, and to a lesser extent in drainage systems and other flooded environments.”
Conclussion
Line 424-425: “Our results reinforce the growing recognition of Cx. perexiguus as a major WNV vector in the Iberian Peninsula, sometimes surpassing Cx. pipiens in abundance. However, the larval data suggest the opposite. Clarification on this point would be appreciated.
The difference in abundance between larvae and adults is surprising. It is possible that the BG-Sentinel traps were positioned near a breeding site of Cx. perexiguus that was not included in the sampling.
Authors: We thank the reviewer for this helpful observation. We agree that the wording in this section was not well expressed and could be misleading. We have modified the text to clarify that while larval data indicated higher numbers of Cx. pipiens within urban foci, adult collections revealed a predominance of Cx. perexiguus. This apparent discrepancy may be explained by the proximity of some BG-Sentinel traps to a larval site of Cx. perexiguus or other unknown reasons. We also acknowledge that the adult results are conditioned by a smaller geographical scale and must therefore be interpreted with caution. This limitation is now explicitly stated in the discussion.
Submission Date
30 August 2025
Date of this review
17 Sep 2025 17:18:42
Round 2
Reviewer 1 Report
Comments and Suggestions for Authors
Message to the Editor
The revised version shows clear progress, with substantial improvements in methodological transparency, data presentation, and narrative focus. Most prior concerns were satisfactorily addressed, and the manuscript now meets the journal’s standards for clarity and scientific rigor.
Remaining issues are minor and primarily stylistic, involving analytical framing and slight verbosity in the Discussion. Overall, I consider the manuscript suitable for publication after minor editorial polishing.
Recommended decision: Minor Revision: Accept after minor edits.
Feedback to Authors
Overall Comments: The revised manuscript, "Rapid-response vector surveillance and emergency control during the largest West Nile Virus outbreak in southern Spain," shows significant improvement compared to the former submission. The revision closely reflects the reviewer's and editor's expectations and acknowledges the effort made by the authors to address the points raised. Particularly, the concerns related to the transparency of the methodology, data reproducibility, and integration of operational and molecular aspects of mosquito surveillance have been substantially addressed.
The work is now more readable, and the claims made are better balanced and supported by quantitative data. However, some parts of the manuscript still need work, primarily those related to the analytical framing of results, the depth of interpretation in the Discussion, and the linking of the descriptive and inferential elements throughout the manuscript. By working on these points, the authors would not only deepen the scientific content of their work but also make it more accessible to a general audience.
Abstract and Simple Summary: The revision successfully eliminates subjective language and presents vital quantitative facts, such as the larval reduction rate (63.9%) and the sampling scale (725 sites, 270,000 residents). The Abstract now also contains useful phylogenetic information on Culex perexiguus and clearly defines the novelty of the study. However, the summary would be clearer if it were framed more statistically, with an explicit mention of the type of model or confidence intervals giving the reader a better understanding of the analytical rigor. Even though the text is brief and factual, it still somewhat describes the study rather than analytically synthesizing it. I would briefly state the statistical approach (e.g., "using generalized linear mixed models") to show that there is analytical depth.
Introduction: The section is now outlining the gap in the research and the novelty of the work more clearly, stressing the combination of operational control with molecular confirmation of Cx. perexiguus as the key point. The newly added paragraph (lines 94-97) is both effective and consistent with the objectives of the study. However, there are still some parts in the text that are too narrative, telling the history of WNV outbreaks in Europe in a way that could be much shorter. By shortening these sentences, the focus would be on the innovative methodological contribution of the study. My suggestion is to shorten background paragraphs so that the focus remains on the operational and molecular integration which is the core of the study’s novelty.
Materials and Methods: The authors have done a great job of moving this section forward. Larvicide and adulticide dosages, reapplication intervals, and equipment types are now very detailed and can be easily checked. It is also very good to see the inclusion of confidence intervals for the follow-up periods and the addition of a statement regarding regulatory compliance. The introduction of a negative binomial GLMM along with clear parameter and post-hoc testing reporting has also significantly improved the section on the statistical framework. A little improvement could be made if the authors included diagnostic information (residual plots or dispersion checks) in the Supplementary Material, thus, they will ensure full reproducibility and transparency of model assumptions. I would insert a short note or supplementary figure showing residual diagnostics and model fit quality.
Results: The section has been considerably reorganized and quantitative. The legends now contain information on sample sizes, p-values, and effect sizes, while the interpretive statements have been suitably relocated to the Discussion. The changed layout enhances the flow of reading and the interpretation of data. Though some redundancy has been left behind, especially the reiteration of numeric values between the text and figure captions, this is only a minor stylistic problem. Be sure not to repeat statistical results in both figure captions and text unless it is necessary for clarity.
Discussion: the revised Discussion demonstrates a significant improvement in profundity and the analytical maturity. It combines ecological, operational, and logistic factors, and the mechanistic explanations are also covered (e.g., Bti degradation, recolonization dynamics, water flow effects). The inclusion of insecticide resistance and adaptive management discussion points is a clear indication of the authors moving forward. However, the manuscript could benefit from more concise phrasing and focusing on synthesis instead of re-explaining. The interpretation of adulticide efficacy is still made in a qualitative way and, therefore, it could be supported by quantitative data or, if not available, by explicitly stating limitations of the evidence base. I suggest that the authors reduce the parts of the text where they explain too much and clarify that conclusions on adulticide performance are mainly based on operational observation, and not on experimental quantification.
Conclusions: the section's mood has been appropriately changed by the authors. The new wording (“Larval control with Bti showed moderate and variable reductions…”) is more accurate in representing the data from the study and it is in agreement with the recommendations for adaptive, habitat-specific control strategies.
Last comment: this is a drastically better manuscript that now provides a coherent, data-driven contribution to operational entomology and public health response. Most of the authors' revisions have resolved the prior concerns. The paper would be at the level of full publication after a final round of stylistic tightening and minor analytical reinforcement.
Author Response
Message to the Editor
The revised version shows clear progress, with substantial improvements in methodological transparency, data presentation, and narrative focus. Most prior concerns were satisfactorily addressed, and the manuscript now meets the journal’s standards for clarity and scientific rigor.
Authors: We sincerely thank the reviewer for their positive and encouraging feedback.
Remaining issues are minor and primarily stylistic, involving analytical framing and slight verbosity in the Discussion. Overall, I consider the manuscript suitable for publication after minor editorial polishing.
Recommended decision: Minor Revision: Accept after minor edits.
Feedback to Authors
Overall Comments: The revised manuscript, "Rapid-response vector surveillance and emergency control during the largest West Nile Virus outbreak in southern Spain," shows significant improvement compared to the former submission. The revision closely reflects the reviewer's and editor's expectations and acknowledges the effort made by the authors to address the points raised. Particularly, the concerns related to the transparency of the methodology, data reproducibility, and integration of operational and molecular aspects of mosquito surveillance have been substantially addressed.
Authors: Thank you.
The work is now more readable, and the claims made are better balanced and supported by quantitative data. However, some parts of the manuscript still need work, primarily those related to the analytical framing of results, the depth of interpretation in the Discussion, and the linking of the descriptive and inferential elements throughout the manuscript. By working on these points, the authors would not only deepen the scientific content of their work but also make it more accessible to a general audience.
Authors: We appreciate the reviewer’s helpful feedback. We agree that further improvements are needed in the analytical framing, interpretation, and integration of results, and we will address these points to enhance the manuscript.
Abstract and Simple Summary: The revision successfully eliminates subjective language and presents vital quantitative facts, such as the larval reduction rate (63.9%) and the sampling scale (725 sites, 270,000 residents). The Abstract now also contains useful phylogenetic information on Culex perexiguus and clearly defines the novelty of the study. However, the summary would be clearer if it were framed more statistically, with an explicit mention of the type of model or confidence intervals giving the reader a better understanding of the analytical rigor. Even though the text is brief and factual, it still somewhat describes the study rather than analytically synthesizing it. I would briefly state the statistical approach (e.g., "using generalized linear mixed models") to show that there is analytical depth.
Authors: We appreciate the reviewer’s valuable suggestion. We agree that a clearer statistical framing would strengthen the Abstract and Simple Summary. However, due to strict word limits and the limited relevance of the statistical results for this section, we could only incorporate a brief reference to the analytical approach while keeping the text concise and focused.
Introduction: The section is now outlining the gap in the research and the novelty of the work more clearly, stressing the combination of operational control with molecular confirmation of Cx. perexiguus as the key point. The newly added paragraph (lines 94-97) is both effective and consistent with the objectives of the study. However, there are still some parts in the text that are too narrative, telling the history of WNV outbreaks in Europe in a way that could be much shorter. By shortening these sentences, the focus would be on the innovative methodological contribution of the study. My suggestion is to shorten background paragraphs so that the focus remains on the operational and molecular integration which is the core of the study’s novelty.
Authors: We appreciate the reviewer’s constructive comments and suggestions. We agree that some background parts were overly narrative, so we have made small adjustments to shorten these sections and keep the focus on the methodological and molecular integration, which represents the main novelty of the study.
Materials and Methods: The authors have done a great job of moving this section forward. Larvicide and adulticide dosages, reapplication intervals, and equipment types are now very detailed and can be easily checked. It is also very good to see the inclusion of confidence intervals for the follow-up periods and the addition of a statement regarding regulatory compliance. The introduction of a negative binomial GLMM along with clear parameter and post-hoc testing reporting has also significantly improved the section on the statistical framework. A little improvement could be made if the authors included diagnostic information (residual plots or dispersion checks) in the Supplementary Material, thus, they will ensure full reproducibility and transparency of model assumptions. I would insert a short note or supplementary figure showing residual diagnostics and model fit quality.
Authors: We thank the reviewer for their positive feedback and helpful suggestion. We agree that including model diagnostics would enhance transparency and reproducibility. Accordingly, we have added a brief note and supplementary figure showing the residual diagnostics and model fit quality.
Results: The section has been considerably reorganized and quantitative. The legends now contain information on sample sizes, p-values, and effect sizes, while the interpretive statements have been suitably relocated to the Discussion. The changed layout enhances the flow of reading and the interpretation of data. Though some redundancy has been left behind, especially the reiteration of numeric values between the text and figure captions, this is only a minor stylistic problem. Be sure not to repeat statistical results in both figure captions and text unless it is necessary for clarity.
Authors: We appreciate the reviewer’s positive assessment and advice. We agree with the observation and have revised the section to minimize redundancy between the text and figure captions, keeping statistical details only where necessary for clarity.
Discussion: the revised Discussion demonstrates a significant improvement in profundity and the analytical maturity. It combines ecological, operational, and logistic factors, and the mechanistic explanations are also covered (e.g., Bti degradation, recolonization dynamics, water flow effects). The inclusion of insecticide resistance and adaptive management discussion points is a clear indication of the authors moving forward.
Authors: Thank you for your positive feedback.
However, the manuscript could benefit from more concise phrasing and focusing on synthesis instead of re-explaining. The interpretation of adulticide efficacy is still made in a qualitative way and, therefore, it could be supported by quantitative data or, if not available, by explicitly stating limitations of the evidence base. I suggest that the authors reduce the parts of the text where they explain too much and clarify that conclusions on adulticide performance are mainly based on operational observation, and not on experimental quantification.
Authors: We appreciate the reviewer’s comment. The text has been revised to make it more concise and focused on synthesis. In addition, we have clarified that the interpretation of adulticide efficacy is based on operational observations rather than quantitative experimental evaluation. The following sentence has been added to the manuscript:
“Although our study did not quantitatively assess the effectiveness of adulticide applications, previous operational experiences, including large-scale interventions in European rice-growing areas, have shown that perimeter spraying with pyrethroid-based products can produce short-term reductions in adult Culex populations. Therefore, conclusions on adulticide performance should be interpreted qualitatively, as they are mainly based on field observations rather than experimental quantification.”
Conclusions: the section's mood has been appropriately changed by the authors. The new wording (“Larval control with Bti showed moderate and variable reductions…”) is more accurate in representing the data from the study and it is in agreement with the recommendations for adaptive, habitat-specific control strategies.
Authors: We thank the reviewer for their positive and encouraging comment. We are glad that the revised conclusion is now considered more accurate and aligned with the study’s findings and recommendations.
Last comment: this is a drastically better manuscript that now provides a coherent, data-driven contribution to operational entomology and public health response. Most of the authors' revisions have resolved the prior concerns. The paper would be at the level of full publication after a final round of stylistic tightening and minor analytical reinforcement.
Authors: We sincerely thank the reviewer for their very positive and encouraging evaluation. We greatly appreciate the constructive feedback throughout the review process, which has helped us substantially improve the quality and clarity of the manuscript.
Reviewer 2 Report
Comments and Suggestions for Authors
I very much support the new focus on WNV as this more clearly connects with the work performed.
There was quite a bit revised and I appreciate the efforts. I did detect just a few minor points to fix.
Line 30: Change “The larvicidal treatments were associated with a 63.9% reduction in larvae and larval abundance declined between 52% and 75% across treatment rounds.” to “The # (?) sequential larvicide treatments were associated with an overall 63.9% reduction that ranged from 52% to 75% across treatment rounds”.
Line 30 and 47: Is it 63.9% or 63.5% (from line 387)?
Line 46: include the number of larvicide treatments
Line 387: Include the number of treatments that were averaged to get 63.5%
Line 264: appears to be a phrase that needs to be deleted
Lines 442, 445, 460, 465 and 476; change “efficacy” to “effectiveness” or “effective control”
Author Response
I very much support the new focus on WNV as this more clearly connects with the work performed.
There was quite a bit revised and I appreciate the efforts. I did detect just a few minor points to fix.
Thank you.
Line 30: Change “The larvicidal treatments were associated with a 63.9% reduction in larvae and larval abundance declined between 52% and 75% across treatment rounds.” to “The # (?) sequential larvicide treatments were associated with an overall 63.9% reduction that ranged from 52% to 75% across treatment rounds”.
Done, thanks.
Line 30 and 47: Is it 63.9% or 63.5% (from line 387)?
Done, thanks.
Line 46: include the number of larvicide treatments
Done, thanks.
Line 387: Include the number of treatments that were averaged to get 63.5%
Done, thanks.
Line 264: appears to be a phrase that needs to be deleted
Done, thanks.
Lines 442, 445, 460, 465 and 476; change “efficacy” to “effectiveness” or “effective control”
Done, thanks.